# CADS: Unleashing the Diversity of Diffusion Models through Condition-Annealed Sampling

**Seyedmorteza Sadat[1], Jakob Buhmann[2], Derek Bradley[2], Otmar Hilliges[1], Romann M. Weber[2]**
[1]ETH Zürich, [2]DisneyResearch|Studios
{seyedmorteza.sadat,otmar.hilliges}@inf.ethz.ch
{jakob.buhmann,derek.bradley,romann.weber}@disneyresearch.com

## Abstract

While conditional diffusion models are known to have good coverage of the data distribution, they still face limitations in output diversity, particularly when sampled with a high classifier-free guidance scale for optimal image quality or when trained on small datasets. We attribute this problem to the role of the conditioning signal in inference and offer an improved sampling strategy for diffusion models that can increase generation diversity, especially at high guidance scales, with minimal loss of sample quality. Our sampling strategy anneals the conditioning signal by adding scheduled, monotonically decreasing Gaussian noise to the conditioning vector during inference to balance diversity and condition alignment. Our Condition-Annealed Diffusion Sampler (CADS) can be used with any pretrained model and sampling algorithm, and we show that it boosts the diversity of diffusion models in various conditional generation tasks. Further, using an existing pretrained diffusion model, CADS achieves a new state-of-the-art FID of 1.70 and 2.31 for class-conditional ImageNet generation at 256×256 and 512×512 respectively.

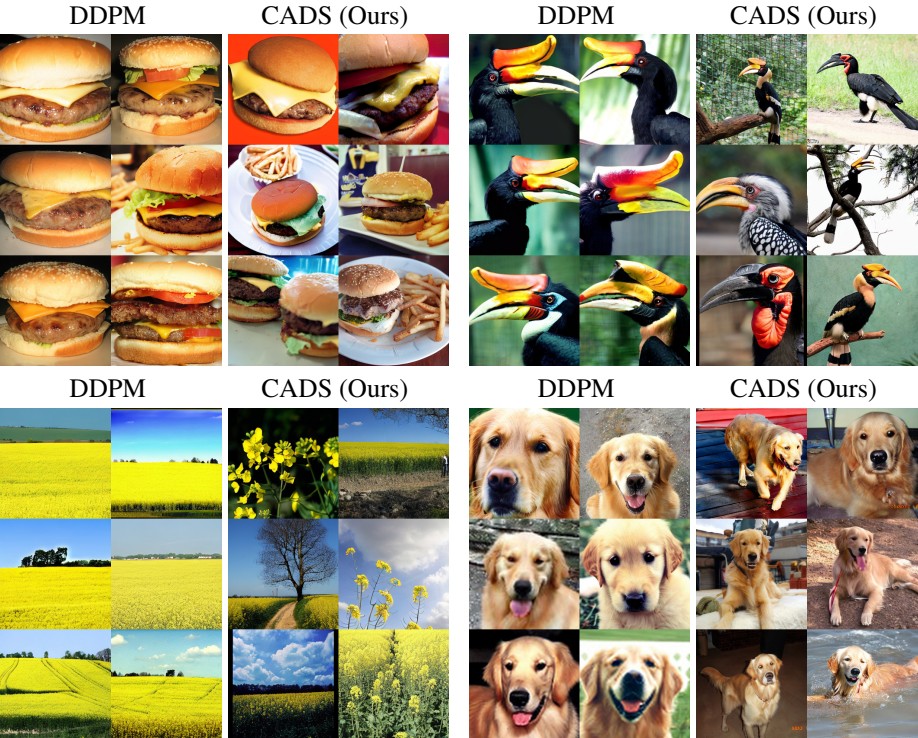

Figure 1: Sampling with classifier-free guidance at high guidance scales using standard methods such as DDPM (Ho et al., 2020) improves image quality but at the cost of *diversity*, leading to sampled images that look similar in composition. We introduce CADS, a sampling technique that significantly increases the diversity of generations while retaining high output quality.

# 1 INTRODUCTION

Generative modeling aims to accurately capture the characteristics of a target data distribution from unstructured inputs, such as images. To be effective, the model must produce diverse results, ensuring comprehensive data distribution coverage, while also generating high-quality samples. Recent advances in generative modeling have elevated denoising diffusion probabilistic models (DDPMs) (Sohl-Dickstein et al., 2015; Ho et al., 2020) to the state of the art in both conditional and unconditional tasks (Dhariwal & Nichol, 2021). Diffusion models stand out not only for their output quality but also for their stability during training, in contrast to Generative Adversarial Networks (GANs) (Goodfellow et al., 2014). They also exhibit superior data distribution coverage, which has been attributed to their likelihood-based training objective (Nichol & Dhariwal, 2021). However, despite considerable progress in refining the architecture and sampling techniques of diffusion models (Dhariwal & Nichol, 2021; Peebles & Xie, 2022; Karras et al., 2022; Hoogeboom et al., 2023; Gao et al., 2023), there has been limited focus on their output diversity.

This paper serves as a comprehensive investigation of the diversity and distribution coverage of conditional diffusion models. We show that conditional diffusion models still suffer from low output diversity when the sampling process uses high classifier-free guidance (CFG) (Ho & Salimans, 2022) or when the underlying diffusion model is trained on a small dataset. One solution would be to reduce the weight of the classifier-free guidance or to train the diffusion model on larger datasets. However, sampling with low classifier-free guidance degrades image quality (Ho & Salimans, 2022; Dhariwal & Nichol, 2021), and collecting a larger dataset may not be feasible in all domains. Even for a diffusion model trained on billions of images, e.g., Stable Diffusion (Rombach et al., 2022), the model may still generate outputs with little variability in response to certain conditions (Figure 4). Thus, merely expanding the dataset does not provide a complete solution to the issue of limited diversity.

Our primary contribution lies in establishing a connection between the conditioning signal and the emergence of low-diversity outputs. We contend that this issue predominantly arises during inference, as a significant majority of samples converge toward the stronger modes within the learned distribution, irrespective of the initial seed. We introduce a simple yet effective technique, termed the Condition-Annealed Diffusion Sampler (CADS), to amplify the diversity of diffusion models. During inference, the conditioning signal is perturbed using additive Gaussian noise combined with an annealing strategy. This strategy introduces significant corruption to the conditioning at the outset of sampling and then gradually reduces the noise to zero by the end. Intuitively, this breaks the statistical dependence on the conditioning signal during early inference, allowing more influence from the data distribution as a whole, and restores that dependence during late inference. As a result, our method can diversify generated samples and simultaneously respect the alignment between the input condition and the generated image.

The principal benefit of CADS is that it does not require any retraining of the underlying diffusion model and can be integrated into all diffusion samplers. Also, its computational overhead is minimal, involving only an additive operation. Through extensive experiments, we show that CADS resolves a long-standing trade-off between the diversity and quality of the output in conditional diffusion models. Moreover, we demonstrate that CADS outperforms the standard DDPM sampling in several conditional generation tasks and sets a new state-of-the-art FID on class-conditional ImageNet (Russakovsky et al., 2015) generation at both 256×256 and 512×512 resolutions while utilizing higher guidance values. Figure 1 showcases the effectiveness of CADS compared to DDPM on the class-conditional ImageNet generation task.

# 2 RELATED WORK

Score-based diffusion models (Song & Ermon, 2019; Song et al., 2021b; Sohl-Dickstein et al., 2015; Ho et al., 2020) learn the data distribution by reversing a forward destruction process that incrementally transforms the data into Gaussian noise. These models quickly surpassed the fidelity and diversity of previous generative modeling schemes (Nichol & Dhariwal, 2021; Dhariwal & Nichol, 2021), achieving state-of-the-art results across domains including unconditional image generation (Dhariwal & Nichol, 2021; Karras et al., 2022), text-to-image generation (Ramesh et al., 2022; Saharia et al., 2022b; Balaji et al., 2022; Rombach et al., 2022), image-to-image translation (Saharia

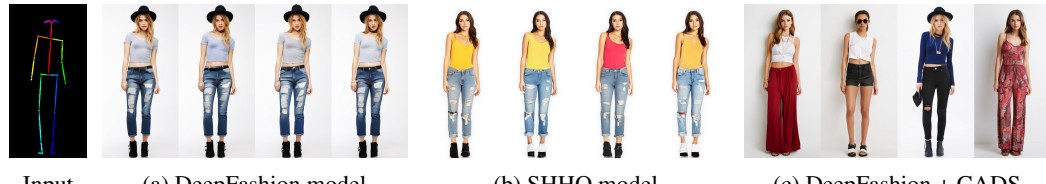

| Input | (a) DeepFashion model | (b) SHHQ model | (c) DeepFashion + CADS |

Figure 2: Low diversity issue in the pose-to-image generation task. (a) The model trained on DeepFashion generates strongly similar outputs. (b) Training on the larger SHHQ dataset only partially solves the issue. (c) Sampling with CADS significantly reduces output similarity.

et al., 2022a; Liu et al., 2023), motion synthesis (Tevet et al., 2023; Tseng et al., 2023), and audio generation (Chen et al., 2021; Kong et al., 2021; Huang et al., 2023).

Building upon the DDPM model (Ho et al., 2020), numerous advancements have been proposed, including architectural refinements (Ho et al., 2020; Hoogeboom et al., 2023; Peebles & Xie, 2022) and improved training techniques (Nichol & Dhariwal, 2021; Karras et al., 2022; Song et al., 2021b; Salimans & Ho, 2022; Rombach et al., 2022). Moreover, different guidance mechanisms (Dhariwal & Nichol, 2021; Ho & Salimans, 2022; Hong et al., 2022; Kim et al., 2023) provided a means to balance sample quality and diversity, reminiscent of truncation tricks in GANs (Brock et al., 2019).

The standard DDPM model employs 1000 sampling steps and can present computational challenges in various applications. As a result, there is an increasing interest in optimizing the sampling efficiency of diffusion models (Song et al., 2021a; Karras et al., 2022; Liu et al., 2022b; Lu et al., 2022a; Salimans & Ho, 2022). These techniques aim to achieve comparable output quality in fewer steps, though they do not necessarily address the diversity issue highlighted in this paper.

Somepalli et al. (2023a) discovered that Stable Diffusion (Rombach et al., 2022) may blindly replicate its training data and offered mitigation strategies in a concurrent work (Somepalli et al., 2023b). In contrast, our work addresses the broader issue of reducing the *similarity* among generated outputs under a given condition, even if they are not direct copies of the training data.

Sehwag et al. (2022) presented a method to sample from low-density regions of the data distribution using a pretrained diffusion model and a likelihood-based hardness score, while our method aims to remain close to the data distribution without specifically targeting low-density regions. Our approach is also compatible with latent diffusion models, unlike the pixel-space focus of Sehwag et al. (2022).

Finally, adding Gaussian noise to the input of a diffusion model is common in cascaded models to avoid the domain gap between generated data from previous stages and real downsampled data (Ho et al., 2021). However, such models require training with noisy inputs, and the noise addition only happens once at the beginning of the inference as opposed to our scheduled approach.

In summary, our work builds on top of recent developments in diffusion models with a direct focus on diversity and can be applied to any architecture and sampling algorithm. Additional background on diffusion models is given in Appendix A for completeness.

## 3    DIVERSITY IN DIFFUSION MODELS

In this paper, *diversity* is defined as the ability of a model to generate varied outputs for a fixed condition when the initial random sample changes. In contrast, models lacking in diversity often generate similar outputs regardless of the random input and tend to focus on a narrower subset of the data distribution.

We trace the low diversity issue in diffusion models to two main factors. First, diffusion models trained on large datasets such as ImageNet (Russakovsky et al., 2015) often require a high classifier-free guidance scale for optimal output quality, but sampling with a high guidance scale is known to reduce diversity (Ho & Salimans, 2022; Murphy, 2023), as shown in Figure 1. Secondly, conditional diffusion models trained on smaller datasets, such as pose-to-image generation on DeepFashion (Liu et al., 2016) ($\approx$ 10k samples) tend to have limited variation, as shown in Figure 2a, even with a low

CFG scale. In both scenarios, the model establishes a near one-to-one mapping from the conditioning signal to the generated images, thereby yielding limited diversity for a given condition.

One potential solution might involve reducing the guidance scale or training the model on larger datasets, like SHHQ (Fu et al., 2022) ($\approx 40$k samples) for the pose-to-image task. However, decreasing the guidance scale compromises image quality (Ho & Salimans, 2022; Dhariwal & Nichol, 2021), and training on larger datasets only partially mitigates the issue, as demonstrated in Figure 2b.

We conjecture that the low-diversity issue is due to a highly peaked conditional distribution, particularly at higher guidance scales, which leads the majority of sampled images toward certain modes. One way to address this issue is by smoothing the conditional distribution with decreasing Gaussian noise, which breaks and then gradually restores the statistical dependency on the conditioning signal during inference. Next, we introduce a novel sampling technique that integrates this intuition into the reverse process of diffusion models. The effectiveness of the method is shown in Figure 2c.

## 3.1 CONDITION-ANNEALED DIFFUSION SAMPLER (CADS)

The core principle of CADS is that by annealing the conditioning signal during inference, the model processes a different input at each step, leading to more diverse outputs. To maintain the essential correspondence between the condition and the output, CADS employs an annealing strategy that reduces the amount of corruption as the inference progresses, causing it to vanish during the final steps of inference. More specifically, inspired by the forward process of diffusion models, we corrupt a given condition $\boldsymbol{y}$ via

$$\hat{\boldsymbol{y}} = \sqrt{\gamma(t)}\boldsymbol{y} + s\sqrt{1 - \gamma(t)}\boldsymbol{n}, \tag{1}$$

where $s$ determines the initial noise scale, $\gamma(t)$ is the annealing schedule, and $\boldsymbol{n} \sim \mathcal{N}(\boldsymbol{0}, \boldsymbol{I})$. This modification can be readily integrated into any sampler, and it significantly diversifies the generations, as demonstrated in Section 4. A discussion on further design choices in CADS follows below.

**Annealing schedule function**  We propose a piecewise linear function for $\gamma(t)$ given by

$$\gamma(t) = \begin{cases} 1 & t \leq \tau_1, \\ \frac{\tau_2 - t}{\tau_2 - \tau_1} & \tau_1 < t < \tau_2, \\ 0 & t \geq \tau_2, \end{cases} \tag{2}$$

for user-defined thresholds $\tau_1, \tau_2 \in [0, 1]$. Since diffusion models operate backward in time from $t = 1$ to $t = 0$ during inference, the annealing function ensures high corruption of $\boldsymbol{y}$ at early steps and no corruption for $t \leq \tau_1$. In Section 4, we show empirically that this choice of $\gamma(t)$ works well in a variety of scenarios.

**Rescaling the conditioning signal**  Adding noise to the condition changes the mean and standard deviation of the conditioning vector. In most experiments, we rescale the conditioning vector back toward its prior mean and standard deviation. Specifically, for a clean condition vector $\boldsymbol{y}$ with (scalar) mean and standard deviation $\mu_{\text{in}}$ and $\sigma_{\text{in}}$, we compute the final corrupted condition $\hat{\boldsymbol{y}}_{\text{final}}$ according to

$$\hat{\boldsymbol{y}}_{\text{rescaled}} = \frac{\hat{\boldsymbol{y}} - \text{mean}(\hat{\boldsymbol{y}})}{\text{std}(\hat{\boldsymbol{y}})}\sigma_{\text{in}} + \mu_{\text{in}} \tag{3}$$

$$\hat{\boldsymbol{y}}_{\text{final}} = \psi\hat{\boldsymbol{y}}_{\text{rescaled}} + (1 - \psi)\hat{\boldsymbol{y}}, \tag{4}$$

for a mixing factor $\psi \in [0, 1]$. This rescaling scheme prevents divergence, especially for high noise scales $s$, but slightly reduces diversity of the outputs. Therefore, one can trade more stable sampling with more diverse generations by changing the mixing factor $\psi$. Section 4.2 contains an ablation study on the effect of rescaling and the mixing factor $\psi$.

## 3.2 INTUITION BEHIND CONDITION ANNEALING

Consider the task of sampling from a conditional diffusion model, where we anneal the condition $\boldsymbol{y}$ according to Equation (1) at each inference step. Using Bayes' rule, the conditional probability at time $t$ can be expressed as $p_t(\boldsymbol{z}_t|\hat{\boldsymbol{y}}) = p_t(\hat{\boldsymbol{y}}|\boldsymbol{z}_t)p_t(\boldsymbol{z}_t)/p(\hat{\boldsymbol{y}})$, and consequently, $\nabla_{\boldsymbol{z}_t} \log p_t(\boldsymbol{z}_t|\hat{\boldsymbol{y}}) = \nabla_{\boldsymbol{z}_t} \log p_t(\hat{\boldsymbol{y}}|\boldsymbol{z}_t) + \nabla_{\boldsymbol{z}_t} \log p_t(\boldsymbol{z}_t)$. When $t$ is close to 1, $\gamma(t) \approx 0$, resulting in $\hat{\boldsymbol{y}} \approx s\boldsymbol{n}$. This indicates that early in the sampling process, $\hat{\boldsymbol{y}}$ is independent of the current noisy sample $\boldsymbol{z}_t$, leading

to $\nabla_{\boldsymbol{z}_t} \log p_t(\hat{\boldsymbol{y}}|\boldsymbol{z}_t) \approx 0$. Hence, the diffusion model initially only follows the unconditional score $\nabla_{\boldsymbol{z}_t} \log p_t(\boldsymbol{z}_t)$ and ignores the condition. As we reduce the noise, the influence of the conditional term increases. This progression ensures more exploration of the space in the early stages and results in high-quality samples with improved diversity. More theoretical insights into condition annealing are detailed in Appendices B and C.

### 3.3 DYNAMIC CLASSIFIER-FREE GUIDANCE

The above analysis motivates us to consider another algorithm to enhance the diversity of diffusion models, one in which we directly modulate the guidance weight to initially *underweight* the contribution of the conditional term. We refer to this method as Dynamic Classifier-Free Guidance. More specifically, in Dynamic CFG, we adjust the guidance weight according to $\hat{w}_{\text{CFG}} = \gamma(t)w_{\text{CFG}}$, forcing the model to rely more on the unconditional score $\nabla_{\boldsymbol{z}_t} \log p_t(\boldsymbol{z}_t)$ early in inference and gradually move toward standard CFG at the end. Section 4.1 details a comparison between Dynamic CFG and CADS. We demonstrate that while Dynamic CFG enhances diversity relative to DDPM, it does not match the performance of CADS. We believe that the additional stochasticity in CADS results in more diverse generations and different sampling dynamics than Dynamic CFG.

## 4 EXPERIMENTS AND RESULTS

In this section, we rigorously evaluate the performance of CADS on various conditional diffusion models and demonstrate that CADS boosts output diversity without compromising quality.

**Setup**   We consider four conditional generation tasks: class-conditional generation on ImageNet (Russakovsky et al., 2015) with DiT-XL/2 (Peebles & Xie, 2022), pose-to-image generation on DeepFashion (Liu et al., 2016) and SHHQ (Fu et al., 2022), identity-conditioned face generation with ID3PM (Kansy et al., 2023), and text-to-image generation with Stable Diffusion (Rombach et al., 2022). We use pretrained networks for all tasks except the pose-to-image generation, in which we train two diffusion models from scratch. DDPM (Ho et al., 2020) is used as the base sampler.

**Sample quality metrics**   We use Fréchet Inception Distance (FID) (Heusel et al., 2017) as the primary metric for capturing both quality and diversity due to its alignment with human judgement. Since FID is sensitive to small implementation details (Parmar et al., 2022), we adopt the evaluation script of ADM (Dhariwal & Nichol, 2021) to ensure a fair comparison with prior work. For completeness, we also report Inception Score (IS) (Salimans et al., 2016) and Precision (Kynkäänniemi et al., 2019). However, it should be noted that IS and Precision cannot accurately evaluate models with diverse outputs since a model producing high-quality but non-diverse samples could artificially achieve high IS and Precision (as shown in Appendix F).

**Diversity metrics**   In addition to FID, we use Recall (Kynkäänniemi et al., 2019) as the main metric for measuring diversity and distribution coverage. Furthermore, we define two additional similarity metrics. Given a set of input conditions, we first compute the pairwise cosine similarity matrix $K_{\boldsymbol{y}}$ among generated images with the same condition, using SSCD (Pizzi et al., 2022) as the pretrained feature extractor. The results are then aggregated for different conditions using two methods: the Mean Similarity Score (MSS), which is a simple average over the similarity matrix $K_{\boldsymbol{y}}$, and the Vendi Score (Friedman & Dieng, 2022), which is based on the Von Neumann entropy of $K_{\boldsymbol{y}}$.

**Implementation details**   For using CADS, we add noise to the class embeddings in DiT-XL/2, face-ID embeddings in ID3PM, and text embeddings in Stable Diffusion. For the pose-to-image generation models, we directly add noise to the pose image. To have a fair comparison among different methods, we use the exact same random seeds for initializing each sampler with and without condition annealing. More implementation details can be found in Appendix G.

### 4.1 MAIN RESULTS

The following sections describe our main findings. Further analyses are provided in Appendix D.

**Qualitative results**   We showcase generated examples using condition annealing alongside the standard DDPM outputs in Figures 1, 3 and 4. These results indicate that CADS increases the

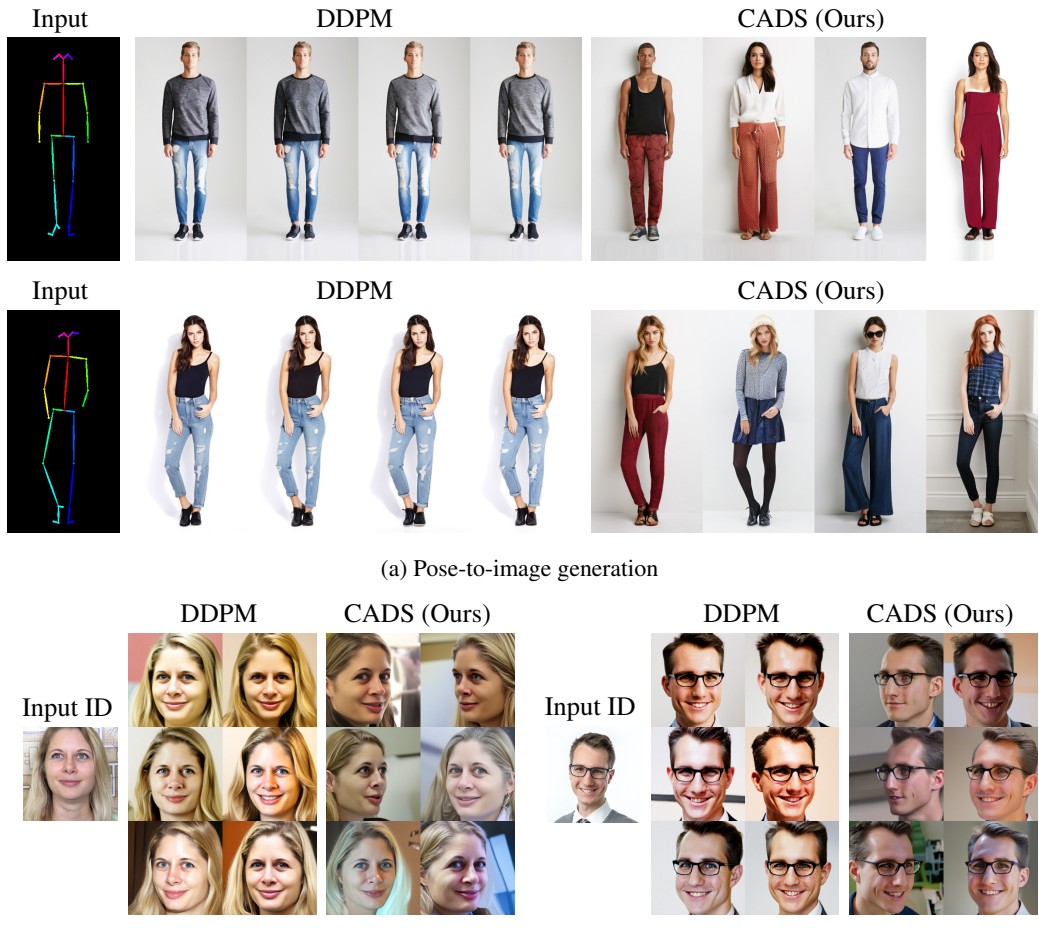

(a) Pose-to-image generation

(b) Identity-conditioned face synthesis

Figure 3: Comparison between DDPM and CADS on two conditional generation tasks. (a) pose-to-image generation based on DeepFashion, and (b) identity-conditioned face synthesis using the ID3PM model (Kansy et al., 2023). In both cases, CADS enhances the diversity of DDPM while maintaining high sample quality.

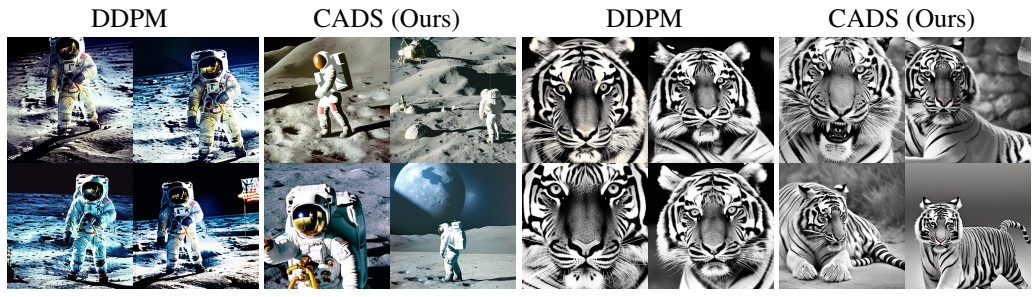

Prompt: An astronaut on the Moon                    Prompt: A grayscale tiger

Figure 4: Two different results from Stable Diffusion v2.1 (Rombach et al., 2022) sampled with DDPM and CADS. CADS enhances the diversity of Stable Diffusion for prompts that yield highly similar images with DDPM.

diversity of the outputs while maintaining high image quality across different tasks. When sampling without annealing and using a high guidance scale, the generated images are realistic but typically lack diversity, regardless of the initial random seed. This issue is especially evident in the DeepFashion model depicted in Figure 3a, where there is an extreme similarity among the generated samples. Interestingly, even the Stable Diffusion model, which usually produces varied samples and is trained on billions of images, can occasionally suffer from low output diversity for specific prompts. As

Table 1: Quantitative comparison between samples generated with DDPM and CADS for a fixed high guidance scale. CADS consistently improves the diversity of the outputs across different tasks as reflected in improved FID, recall, and similarity scores.

| Dataset | Sampler | FID ↓ | Precision ↑ | Recall ↑ | MSS ↓ | Vendi Score ↑ |
|---|---|---|---|---|---|---|
| DeepFashion ($w_{CFG} = 4$) | DDPM | 16.36 | **0.90** | 0.02 | 0.80 | 1.04 |
| | CADS (Ours) | **7.73** | 0.77 | **0.48** | **0.30** | **2.31** |
| SHHQ ($w_{CFG} = 4$) | DDPM | 17.93 | **0.74** | 0.17 | 0.61 | 1.22 |
| | CADS (Ours) | **10.37** | 0.65 | **0.43** | **0.44** | **1.65** |
| ImageNet 256 ($w_{CFG} = 5$) | DDPM | 20.83 | **0.92** | 0.32 | 0.19 | 5.07 |
| | CADS (Ours) | **9.47** | 0.82 | **0.62** | **0.08** | **7.98** |
| ImageNet 512 ($w_{CFG} = 5$) | DDPM | 23.10 | **0.82** | 0.28 | 0.20 | 4.88 |
| | CADS (Ours) | **9.81** | **0.82** | **0.52** | **0.09** | **7.75** |
| ID3PM ($w_{CFG} = 4$) | DDPM | 16.33 | 0.65 | 0.26 | 0.44 | 1.71 |
| | CADS (Ours) | **11.86** | **0.67** | **0.31** | **0.34** | **2.44** |
| Stable Diffusion ($w_{CFG} = 9$) | DDPM | 49.48 | **0.67** | 0.25 | 0.21 | 4.80 |
| | CADS (Ours) | **44.16** | 0.63 | **0.37** | **0.15** | **6.41** |

Table 2: Benchmark for class-conditional generation on ImageNet 256×256 and 512×512. Sampling with CADS improves the FID of DiT-XL/2 to the state-of-the-art at both resolutions while using a higher guidance value and without any retraining of the underlying diffusion model.

| Model | ImageNet 256×256 | | | | ImageNet 512×512 | | | |
|---|---|---|---|---|---|---|---|---|
| | FID ↓ | IS ↑ | Precision ↑ | Recall ↑ | FID ↓ | IS ↑ | Precision ↑ | Recall ↑ |
| BigGAN-deep (Brock et al., 2019) | 6.95 | 171.40 | **0.87** | 0.28 | 8.43 | 177.90 | **0.88** | 0.29 |
| StyleGAN-XL (Sauer et al., 2022) | 2.30 | 265.12 | 0.78 | 0.53 | 2.41 | 267.75 | 0.77 | 0.52 |
| ADM-G, ADM-U (Dhariwal & Nichol, 2021) | 3.94 | 215.84 | 0.83 | 0.53 | 3.85 | 221.72 | 0.84 | 0.53 |
| LDM-4-G ($w_{CFG} = 1.5$) Rombach et al. (2022) | 3.60 | 247.67 | **0.87** | 0.48 | - | - | - | - |
| RIN+NoiseSchedule (Chen, 2023) | 3.52 | 186.20 | - | - | 3.95 | 216.00 | - | - |
| SimpleDiffusion (Hoogeboom et al., 2023) | 2.44 | 256.30 | - | - | 3.02 | 248.70 | - | - |
| VDM++ (Kingma & Gao, 2023) | 2.12 | 267.70 | - | - | 2.65 | **278.10** | - | - |
| DiT-G++ (Kim et al., 2023) | 1.83 | 281.53 | 0.78 | **0.64** | - | - | - | - |
| MDT-G (Gao et al., 2023) | 1.79 | 283.01 | 0.81 | 0.61 | - | - | - | - |
| DiT-XL/2-G ($w_{CFG} = 1.5$) (Peebles & Xie, 2022) | 2.27 | 278.24 | 0.83 | 0.57 | 3.04 | 240.82 | 0.84 | 0.54 |
| DiT-XL/2 with CADS ($w_{CFG} = 2$) | **1.70** | 268.77 | 0.77 | **0.64** | 2.53 | 219.08 | 0.80 | **0.61** |
| DiT-XL/2 with CADS ($w_{CFG} = 2.25$) | 1.81 | 288.10 | 0.78 | **0.64** | 2.32 | 233.03 | 0.80 | 0.60 |
| DiT-XL/2 with CADS ($w_{CFG} = 2.5$) | 1.93 | **297.96** | 0.78 | 0.63 | **2.31** | 239.56 | 0.80 | **0.61** |

demonstrated in Figure 4, CADS successfully addresses this problem. More visual results are provided in Appendix H.

**Quantitative evaluation**  Table 1 quantitatively compares CADS with DDPM across various tasks for a fixed high guidance scale. We observe that CADS significantly enhances the diversity of samples, leading to consistent improvements in FID, Recall, and similarity scores for all tasks. This enhanced performance comes without a significant drop in Precision, indicating that CADS increases generation diversity while maintaining high-quality outputs. As expected, the benefits of CADS is less pronounced in Stable Diffusion since the model is already capable of diverse generations.

**State-of-the-art ImageNet generation**  By employing higher guidance values while maintaining diversity, CADS achieves a new state-of-the-art FID of 1.70 for class-conditional generation on ImageNet 256×256, as illustrated in Table 2. Remarkably, CADS surpasses the previous best FID of MDT (Gao et al., 2023) solely through improved sampling and without the need for retraining the diffusion model. Our approach also sets a new state-of-the-art FID of 2.31 for class-conditional generation on ImageNet 512×512, reinforcing the effectiveness of CADS.

**CADS vs guidance scale**  Classifier-free guidance has been demonstrated to enhance the quality of generated images at the cost of diminished diversity, particularly at higher guidance scales. In this experiment, we illustrate how CADS can substantially alleviate this trade-off. Figure 5 supports this finding. With condition annealing, both FID and Recall exhibit less drastic deterioration as the

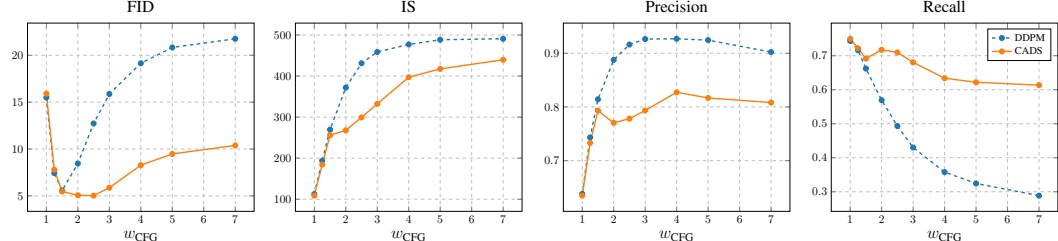

Figure 5: The behavior of the evaluation metrics across different guidance scales. CADS exhibits superior ability to balance quality and diversity, evidenced by better performance in FID and Recall.

Table 3: Impact of integrating CADS with popular diffusion samplers using the class-conditional ImageNet model (DiT-XL/2). CADS enhances sample diversity across all samplers.

| | with CADS | | without CADS | |
|---|---|---|---|---|
| Sampler | FID ↓ | Recall ↑ | FID ↓ | Recall ↑ |
| DDIM (Song et al., 2021a) | **9.80** | **0.59** | 18.84 | 0.35 |
| DPM++ (Lu et al., 2022b) | **9.63** | **0.61** | 18.65 | 0.36 |
| SDE-DPM++ (Lu et al., 2022b) | **11.25** | **0.56** | 19.49 | 0.35 |
| PNDM (Liu et al., 2022a) | **14.60** | **0.52** | 20.23 | 0.32 |
| UniPC (Zhao et al., 2023) | **10.10** | **0.59** | 18.90 | 0.35 |

guidance scale increases, contrasting with the behavior observed in DDPM. Note that because higher guidance values lead to lower diversity, we increase the amount of noise used in CADS relative to the guidance scale to fully leverage the potential of CADS. This is done by lowering $\tau_1$ and increasing $s$ for higher $w_{CFG}$ values. As we increase $w_{CFG}$, DDPM's diversity becomes more limited, and the gap between CADS and DDPM expands. Thus, CADS unlocks the benefits of higher guidance scales without a significant decline in output diversity.

**Different diffusion samplers** Table 3 demonstrates that CADS is compatible with common off-the-shelf diffusion model samplers. Similar to previous experiments, incorporating CADS into each sampler consistently improves output diversity, as indicated by considerably better FID and Recall.

**Dynamic CFG vs CADS** We next compare DDPM sampling with CADS and Dynamic CFG. Table 4 indicates that although both CADS and Dynamic CFG increase the output diversity of DDPM in the class-conditional model, CADS generally leads to more diverse outputs and outperforms Dynamic CFG in the FID score. This is also reflected in the recall values of $0.62$ for CADS and $0.39$ for Dynamic CFG. Hence, we contend that CADS per-

Table 4: Comparison between CADS and Dynamic CFG on class-conditional ImageNet generation.

| Sampler | FID ↓ | Recall ↑ |
|---|---|---|
| DDPM | 20.83 | 0.32 |
| Dynamic CFG | 18.42 | 0.39 |
| CADS | **9.47** | **0.62** |

forms better than simply modulating the guidance weight during inference. Figure 6 also showcases the general similarity between generated samples based on CADS and Dynamic CFG, confirming the theoretical intuition introduced in Section 3.2.

**Condition alignment** To examine the impact of noise on the alignment between the input condition and the generated image, we use three metrics: top-1 accuracy for class-conditional ImageNet generation, mean-per-joint pose error (MPJPE) for the pose-to-image model, and CLIP-score for text-to-image synthesis. Results in Table 5 indicate that pose and text alignments are identical between DDPM and CADS. For class-conditional

Table 5: Condition alignment of different methods after using CADS.

| Alignment metric | DDPM | CADS |
|---|---|---|
| Top-1 Class Accuracy ↑ | 0.98 | 0.96 |
| MPJPE ↓ | 0.02 | 0.02 |
| CLIP-Score ↑ | 0.31 | 0.31 |

generation, only a minor accuracy decline is observed, likely due to increased output variation. Hence, condition annealing does not compromise the adherence to the input conditions.

## 4.2 ABLATION STUDIES

This section explores the role of the most important parameters in CADS on the final quality and diversity of generated samples. Additional ablations on the role of $\tau_2$, the distribution of $\boldsymbol{n}$, and the functional form of $\gamma(t)$ are provided in Appendix E.

DDPM                    Dynamic CFG                    CADS

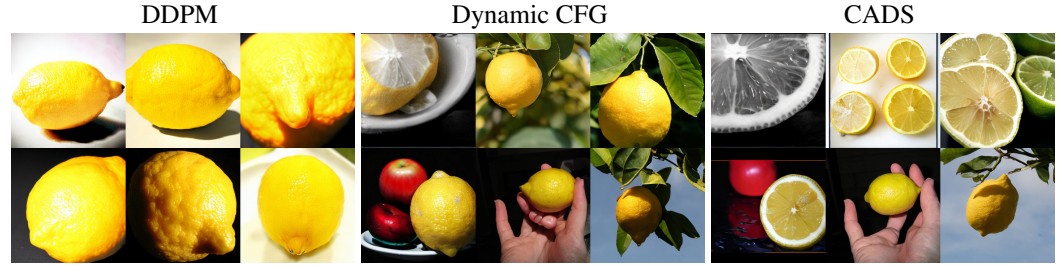

Figure 6: A comparison between condition annealing (CADS) and Dynamic CFG. Both can effectively increase diversity in generated outputs, but CADS generally offers more variations.

Table 6: Ablation study examining various design elements in CADS.

(a) Influence of the noise scale $s$    (b) Impact of the cut-off threshold $\tau_1$    (c) Effect of the mixing factor $\psi$

| $s$ | FID $\downarrow$ | Recall $\uparrow$ |
|-----|------|--------|
| 0.025 | 19.51 | 0.35 |
| 0.1 | **15.79** | 0.52 |
| 0.25 | 42.58 | **0.69** |

| $\tau_1$ | FID $\downarrow$ | Recall $\uparrow$ |
|-----|------|--------|
| 0.2 | 69.73 | **0.80** |
| 0.6 | **15.79** | 0.52 |
| 0.9 | 20.71 | 0.32 |

| $\psi$ | FID $\downarrow$ | Recall $\uparrow$ |
|-----|------|--------|
| 0 | 85.25 | **0.78** |
| 0.5 | 13.48 | 0.56 |
| 1 | **12.18** | 0.55 |

$s = 0.025$     $s = 0.25$     $\tau_1 = 0.2$     $\tau_1 = 0.9$     $\psi = 0.0$     $\psi = 1.0$

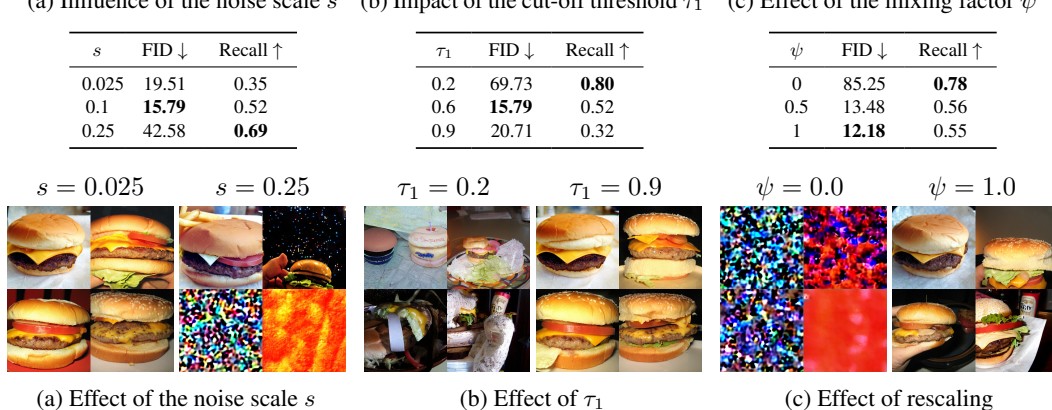

(a) Effect of the noise scale $s$      (b) Effect of $\tau_1$      (c) Effect of rescaling

Figure 7: Visual illustration of how different hyperparameters affect CADS.

**Noise scale** $s$    The effect of the noise scale $s$ on the generated images is illustrated in Figure 7a and Table 6a. Introducing minimal noise ($s = 0.025$) to the conditioning results in reduced diversity, while excessive noise ($s = 0.25$) adversely affects the quality of the samples.

**Cut-off threshold** $\tau_1$    Table 6b and Figure 7b show the influence of the annealing cut-off point $\tau_1$ on the final samples. Analogous to the noise scale, too much noise ($\tau_1 = 0.2$) degrades image quality, while too little noise ($\tau_1 = 0.9$) results in minimal improvement in diversity.

**Effect of rescaling**    Rescaling serves as a regularizer that controls the total amount of noise injected into the condition and helps prevent divergence, especially when the noise scale $s$ is high. This effect is illustrated in Figure 7c. Table 6c also reveals that an increase in $\psi$ (indicating more regularization) typically results in higher sample quality (improved FID) but reduces diversity (lower Recall). We recommend using CADS with $\psi = 1$ and decreasing it only if the attained diversity is insufficient.

## 5    CONCLUSION AND DISCUSSION

In this work, we studied the long-standing challenge of diversity-quality trade-off in diffusion models and showed that the quality benefits of high classifier-free guidance scales can be maintained without sacrificing diversity. This is achieved by controlling the dependence on the conditioning signal via the addition of decreasing levels of Gaussian noise during inference. Our method, CADS, does not require any retraining of pretrained models and can be applied on top of all diffusion samplers. Our experiments confirmed that CADS leads to high-quality, diverse outputs and sets a new benchmark in FID score for class-conditional ImageNet generation. Further, we showed that CADS outperforms the naïve approach of selectively underweighting the guidance scale during inference. Challenges remain for applying CADS to broader conditioning contexts, such as segmentation maps with dense spatial semantics, which we consider as promising avenue for future work.

ETHICS STATEMENT

With the advancement of generative modeling, the ease of creating and disseminating fabricated or inaccurate data increases. Consequently, while progress in AI-generated content can enhance productivity and creativity, careful consideration is essential due to the associated risks and ethical implications. We refer the readers to Rostamzadeh et al. (2021) for a more thorough treatment of ethics and creativity in computer vision.

REPRODUCIBILITY STATEMENT

Our work builds upon publicly available datasets and the official implementations of the pretrained models cited in the main text. The precise algorithms for CADS and Dynamic CFG are detailed in Algorithms 1 and 2, along with the pseudocode for the annealing schedule and noise addition in Figure 15. Further implementation details as well as the exact hyperparameters used to obtain the results presented in this paper are outlined in Appendix G.

ACKNOWLEDGMENT

We would like to thank ETH Zürich Euler computing cluster for providing the infrastructure to run our experiments. Moreover, we thank Xu Chen and Manuel Kansy for helpful discussions during the development of the project and helping out with the final version of the manuscript.

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

# A  BACKGROUND ON DIFFUSION MODELS

We provide an overview of diffusion models in this section for completeness. Please refer to Yang et al. (2022) and Karras et al. (2022) for a more comprehensive treatment of the subject.

Let $\boldsymbol{x} \sim p_{\text{data}}(\boldsymbol{x})$ be a data point, $t \in [0, 1]$ be the time step, and $\boldsymbol{z}_t = \boldsymbol{x} + \sigma(t)\boldsymbol{\varepsilon}$ be the forward process that adds Gaussian noise $\boldsymbol{\varepsilon} \sim \mathcal{N}(\mathbf{0}, \boldsymbol{I})$ to the data for a monotonically increasing function $\sigma(t)$ satisfying $\sigma(0) = 0$ and $\sigma(1) = \sigma_{\max} \gg \sigma_{\text{data}}$. Karras et al. (2022) showed that the evolution of the noisy samples $\boldsymbol{z}_t$ can be described by the ordinary differential equation (ODE)

$$\mathrm{d}\boldsymbol{z} = -\dot{\sigma}(t)\sigma(t)\,\nabla_{\boldsymbol{z}_t}\log p_t(\boldsymbol{z}_t)\,\,\mathrm{d}t, \tag{5}$$

or equivalently, the stochastic differential equation (SDE)

$$\mathrm{d}\boldsymbol{z} = -\dot{\sigma}(t)\sigma(t)\,\nabla_{\boldsymbol{z}_t}\log p_t(\boldsymbol{z}_t)\,\mathrm{d}t - \beta(t)\sigma(t)^2\,\nabla_{\boldsymbol{z}_t}\log p_t(\boldsymbol{z}_t) + \sqrt{2\beta(t)}\sigma(t)\,\mathrm{d}\omega_t. \tag{6}$$

Here, $\mathrm{d}\omega_t$ is the standard Wiener process, and $p_t(\boldsymbol{z}_t)$ is the distribution of perturbed samples with $p_0 = p_{\text{data}}$ and $p_1 = \mathcal{N}(\mathbf{0}, \sigma_{\max}^2\boldsymbol{I})$. Assuming we have access to the time-dependent score function $\nabla_{\boldsymbol{z}_t}\log p_t(\boldsymbol{z}_t)$, sampling from diffusion models is equivalent to solving the diffusion ODE or SDE backward in time from $t = 1$ to $t = 0$. The unknown score function $\nabla_{\boldsymbol{z}_t}\log p_t(\boldsymbol{z}_t)$ is approximated using a parameterized denoiser $D_{\boldsymbol{\theta}}(\boldsymbol{z}_t, t)$ trained on predicting the clean samples $\boldsymbol{x}$ from the corresponding noisy samples $\boldsymbol{z}_t$. The framework allows for conditional generation by training a denoiser $D_{\boldsymbol{\theta}}(\boldsymbol{x}, t, \boldsymbol{y})$ that accepts additional input signals $\boldsymbol{y}$, such as class labels or text prompts.

**Variance-preserving (VP) formulation**  VP is another way to define the forward process of diffusion models. Note that the process $\boldsymbol{z}_t = \boldsymbol{x} + \sigma(t)\boldsymbol{\varepsilon}$ has time-dependent scale governed by $\sigma(t)$. In contrast, the VP formulation destroys the signal $\boldsymbol{x}$ using $\boldsymbol{z}_t = \alpha_t\boldsymbol{x} + \sigma_t\boldsymbol{\varepsilon}$. The parameters $\alpha_t$ and $\sigma_t$ determine how the original signal $\boldsymbol{x}$ is degraded over time, with $\sigma_t = \sqrt{1 - \alpha_t^2}$ being a common choice for $\alpha_t \in [0, 1]$. This forward process ensures that the variance of the noisy samples $\boldsymbol{z}_t$ is better constrained over time.

**Training objective**  Given a noisy sample $\boldsymbol{z}_t$, the denoiser $D_{\boldsymbol{\theta}}(\boldsymbol{z}_t, t, \boldsymbol{y})$ with parameters $\boldsymbol{\theta}$ can be trained with the standard MSE loss

$$\arg\min_{\boldsymbol{\theta}} \mathbb{E}_{t \sim \mathcal{U}(0,1)}\Big[\big\|D_{\boldsymbol{\theta}}(\boldsymbol{z}_t, t, \boldsymbol{y}) - \boldsymbol{x}\big\|^2\Big]. \tag{7}$$

It is also common to formulate the denoiser as a network $\boldsymbol{\varepsilon}_{\boldsymbol{\theta}}(\boldsymbol{z}_t, t, \boldsymbol{y})$ that predicts the total added noise $\boldsymbol{\varepsilon}$ instead of the clean signal $\boldsymbol{x}$. The training objective for diffusion models in this case is

$$\arg\min_{\boldsymbol{\theta}} \mathbb{E}_{t \sim \mathcal{U}(0,1)}\Big[\big\|\boldsymbol{\varepsilon}_{\boldsymbol{\theta}}(\boldsymbol{z}_t, t, \boldsymbol{y}) - \boldsymbol{\varepsilon}\big\|^2\Big]. \tag{8}$$

Recently, Salimans & Ho (2022) found that predicting the "velocity" $\boldsymbol{v}_t = \alpha_t\boldsymbol{\varepsilon} - \sigma_t\boldsymbol{x}$ instead of the noise $\boldsymbol{\varepsilon}$ results in more stable training and faster convergence, similar to the preconditioning step introduced by Karras et al. (2022). After training, the denoiser approximates the time-dependent score function $\nabla_{\boldsymbol{z}_t}\log p_t(\boldsymbol{z}_t)$ via

$$\nabla_{\boldsymbol{z}_t}\log p_t(\boldsymbol{z}_t) \approx \frac{\boldsymbol{x} - D_{\boldsymbol{\theta}}(\boldsymbol{z}_t, t, \boldsymbol{y})}{\sigma(t)} \approx -\frac{\boldsymbol{\varepsilon}_{\boldsymbol{\theta}}(\boldsymbol{z}_t, t, \boldsymbol{y})}{\sigma(t)}. \tag{9}$$

**Latent diffusion models (LDMs)**  LDMs (Rombach et al., 2022) are similar to image-space diffusion models but operate on the latent space of a frozen autoencoder. Given an encoder $\mathcal{E}$ and a decoder $\mathcal{D}$ trained purely with reconstruction loss, LDMs first convert all training samples $\boldsymbol{x}$ to their corresponding latent vectors $\tilde{\boldsymbol{x}} = \mathcal{E}(\boldsymbol{x})$ and train the diffusion model on $\tilde{\boldsymbol{x}}$. We then sample new latents $\tilde{\boldsymbol{x}}$ from the diffusion model and get the corresponding data through the decoder, with $\boldsymbol{x} = \mathcal{D}(\tilde{\boldsymbol{x}})$.

**Classifier-free guidance (CFG)**  CFG is a method for controlling the quality of generated outputs at inference by mixing the predictions of a conditional and an unconditional model (Ho & Salimans, 2022). Specifically, given a null condition $\boldsymbol{y}_{\text{null}} = \varnothing$ corresponding to the unconditional case, CFG modifies the output of the denoiser at each step based on

$$\hat{D}_{\boldsymbol{\theta}}(\boldsymbol{z}_t, t, \boldsymbol{y}) = D_{\boldsymbol{\theta}}(\boldsymbol{z}_t, t, \boldsymbol{y}_{\text{null}}) + w_{\text{CFG}}(D_{\boldsymbol{\theta}}(\boldsymbol{z}_t, t, \boldsymbol{y}) - D_{\boldsymbol{\theta}}(\boldsymbol{z}_t, t, \boldsymbol{y}_{\text{null}})), \tag{10}$$

where $w_{\text{CFG}} = 1$ corresponds to the non-guided case. The unconditional model $D_{\boldsymbol{\theta}}(\boldsymbol{z}_t, t, \boldsymbol{y}_{\text{null}})$ is trained by randomly assigning the null condition $\boldsymbol{y}_{\text{null}} = \varnothing$ to the input of the denoiser for a portion of training. Similar to the truncation method in GANs (Brock et al., 2019), CFG increases the quality of individual images at the expense of less diversity (Murphy, 2023).

## B    CONDITION ANNEALING AND LANGEVIN DYNAMICS

The purpose of this section is to show the connection between condition annealing and Langevin dynamics. As stated in Appendix A, sampling from diffusion models can be formulated as solving the diffusion ODE given by Equation (5). One straightforward solution to this ODE is through Euler's method: $\boldsymbol{z}_{t-1} = \boldsymbol{z}_t + \rho_t \nabla_{\boldsymbol{z}_t} \log p(\boldsymbol{z}_t)$ for some step size $\rho_t$. Note that the denoiser in diffusion models provides an approximation for the score function (Karras et al., 2022). That is, for a given condition $\boldsymbol{y}$, we have

$$\nabla_{\boldsymbol{z}_t} \log p(\boldsymbol{z}_t|\boldsymbol{y}) \approx \frac{\boldsymbol{x} - D(\boldsymbol{z}_t, t, \boldsymbol{y})}{\sigma(t)}. \tag{11}$$

For simplicity, assume that we corrupt the condition according to $\hat{\boldsymbol{y}} = \boldsymbol{y} + \sigma_c \boldsymbol{n}$, where the noise term $\sigma_c \boldsymbol{n}$ is small compared to $\boldsymbol{y}$. Based on the Taylor expansion, we can approximate the output of the denoiser as $D(\boldsymbol{z}_t, t, \hat{\boldsymbol{y}}) \approx D(\boldsymbol{z}_t, t, \boldsymbol{y}) + \sigma_c \nabla_{\boldsymbol{y}} D(\boldsymbol{z}_t, t, \boldsymbol{y})^\top \boldsymbol{n}$, and

$$\boldsymbol{z}_{t-1} \approx \boldsymbol{z}_t + \rho_t \nabla_{\boldsymbol{z}_t} \log p(\boldsymbol{z}_t|\boldsymbol{y}) - \frac{\rho_t}{\sigma(t)} \sigma_c \nabla_{\boldsymbol{y}} D(\boldsymbol{z}_t, t, \boldsymbol{y})^\top \boldsymbol{n}. \tag{12}$$

Let $\boldsymbol{\eta}_t = -\frac{\rho_t}{\sigma(t)} \sigma_c \nabla_{\boldsymbol{y}} D(\boldsymbol{z}_t, t, \boldsymbol{y})$. The update rule in Equation (12) is then equivalent to

$$\boldsymbol{z}_{t-1} \approx \boldsymbol{z}_t + \rho_t \nabla_{\boldsymbol{z}_t} \log p(\boldsymbol{z}_t|\boldsymbol{y}) + \boldsymbol{\eta}_t^\top \boldsymbol{n}. \tag{13}$$

Hence, incorporating noise into the condition is largely analogous to a Langevin dynamics step, albeit with potentially non-isotropic additive Gaussian noise, guided by the underlying diffusion ODE. This parallels conventional Langevin dynamics, where noise addition mitigates mode collapse and improves data distribution coverage. It should be noted that ancestral sampling algorithms like DDPM already incorporate a noise addition step in the reverse process. However, our empirical results in Section 4.1 indicate that this step alone is insufficient to ensure output diversity. Conversely, the noise introduced in condition-annealing, governed by $\boldsymbol{\eta}_t$, appears to provide more effective sampling dynamics.

## C    CONDITION ANNEALING AS SCORE SMOOTHING

In this section, we provide a rough theoretical analysis of CADS in a manner similar to Song et al. (2023), which is based on Tweedie's formula for a simple case in which the condition is linearly related to the input, i.e., $\boldsymbol{y} = \mathbf{T}\boldsymbol{x}$. Note that diffusion models sample from the conditional distribution by following the score $\nabla_{\boldsymbol{z}_t} \log p_t(\boldsymbol{z}_t|\boldsymbol{y})$ at each time step. According to Bayes' rule, we have

$$\nabla_{\boldsymbol{z}_t} \log p_t(\boldsymbol{z}_t|\boldsymbol{y}) = \nabla_{\boldsymbol{z}_t} \log p_t(\boldsymbol{z}_t) + \nabla_{\boldsymbol{z}_t} \log p_t(\boldsymbol{y}|\boldsymbol{z}_t). \tag{14}$$

As the first term does not depend on $\boldsymbol{y}$, we analyze what happens to the second term when we add noise to $\boldsymbol{y}$ via $\hat{\boldsymbol{y}} = \boldsymbol{y} + \sigma_c \boldsymbol{n}$, where $\sigma_c$ controls the amount of noise at each step. That is, we aim to find an approximation for the conditional score $\nabla_{\boldsymbol{z}_t} \log p_t(\hat{\boldsymbol{y}}|\boldsymbol{z}_t)$. First, note that we have

$$p_t(\boldsymbol{y}|\boldsymbol{z}_t) = \int_{\boldsymbol{x}} p_t(\boldsymbol{y}|\boldsymbol{x}) p_t(\boldsymbol{x}|\boldsymbol{z}_t) \, d\boldsymbol{x}. \tag{15}$$

The distribution $p_t(\boldsymbol{x}|\boldsymbol{z}_t)$ is intractable, so we approximate it by a Gaussian distribution around the mean $\hat{\boldsymbol{x}}_t$. That is, we assume $p_t(\boldsymbol{x}|\boldsymbol{z}_t) \approx \mathcal{N}(\hat{\boldsymbol{x}}_t, \lambda \mathbf{I})$ for some $\lambda$. If the forward process for the diffusion model is given by $\boldsymbol{z}_t = \boldsymbol{x} + \sigma(t)\boldsymbol{\varepsilon}$, the mean $\hat{\boldsymbol{x}}_t$ of the distribution $p_t(\boldsymbol{x}|\boldsymbol{z}_t)$ can be estimated based on Tweedie's formula, namely $\hat{\boldsymbol{x}}_t = \mathbb{E}[\boldsymbol{x}|\boldsymbol{z}_t] = \boldsymbol{z}_t + \sigma(t) \nabla_{\boldsymbol{z}_t} \log p_t(\boldsymbol{z}_t)$. By carrying the Gaussian approximation forward and applying the transformation $\mathbf{T}$, we obtain $p_t(\boldsymbol{y}|\boldsymbol{z}_t) \approx \mathcal{N}(\mathbf{T}\hat{\boldsymbol{x}}_t, \lambda^2 \mathbf{T}\mathbf{T}^\top)$. This means we can write $\boldsymbol{y}|\boldsymbol{z}_t = \mathbf{T}\hat{\boldsymbol{x}}_t + \lambda \mathbf{T}\boldsymbol{u}$ for a random variable $\boldsymbol{u} \sim \mathcal{N}(\mathbf{0}, \mathbf{I})$. Hence, the annealed condition is given by $\hat{\boldsymbol{y}}|\boldsymbol{z}_t = \mathbf{T}\hat{\boldsymbol{x}}_t + \lambda \mathbf{T}\boldsymbol{u} + \sigma_c \boldsymbol{n}$, and we have $p_t(\hat{\boldsymbol{y}}|\boldsymbol{z}_t) = \mathcal{N}(\mathbf{T}\hat{\boldsymbol{x}}_t, \lambda^2 \mathbf{T}\mathbf{T}^\top + \sigma_c^2 \mathbf{I})$. This leads to

$$\nabla_{\boldsymbol{z}_t} \log p_t(\hat{\boldsymbol{y}}|\boldsymbol{z}_t) = (\hat{\boldsymbol{y}} - \mathbf{T}\hat{\boldsymbol{x}}_t)^\top (\lambda^2 \mathbf{T}\mathbf{T}^\top + \sigma_c^2 \mathbf{I})^{-1} \mathbf{T} \frac{\partial \hat{\boldsymbol{x}}_t}{\partial \boldsymbol{z}_t}. \tag{16}$$

Compared to the noiseless case, $\sigma_c = 0$, adding noise smooths the inverse term $(\lambda^2 \mathbf{T}\mathbf{T}^\top + \sigma_c^2 \mathbf{I})^{-1}$, similar to the effect of $\ell_2$ regularization in ridge regression. This regularization should on average reduce $\|\nabla_{\boldsymbol{z}_t} \log p_t(\hat{\boldsymbol{y}}|\boldsymbol{z}_t)\|$ and, hence, increase the diversity of generations by preventing the domination of strong modes in the inference process. A similar phenomenon was observed empirically in Dhariwal & Nichol (2021).

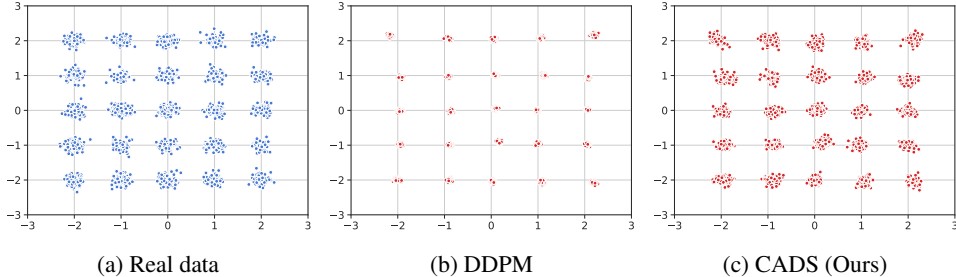

Figure 8: Effect of condition annealing on a toy problem. The real samples from the data distribution are shown in (a). When sampling with high guidance values, standard DDPM converges to the mode of each component (b), but sampling with CADS results in better coverage of each mode (c).

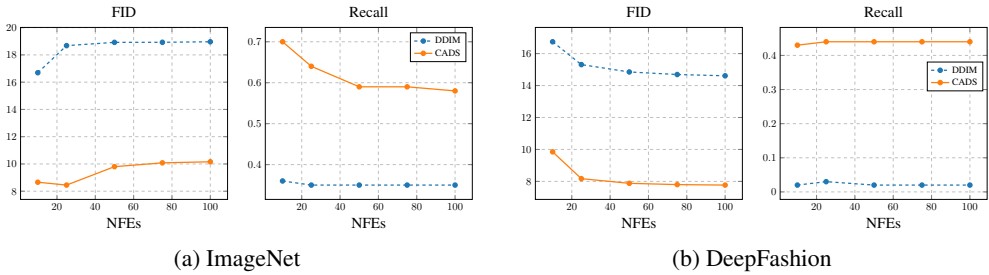

Figure 9: The behavior of CADS versus different NFEs for the class-conditional (ImageNet) and pose-to-image (DeepFashion) models. CADS exhibits similar behavior to the standard sampler as the NFE increases, while consistently outperforming DDIM at each NFE.

## D  ADDITIONAL EXPERIMENTS

We provide additional experiments in this section to extend our understanding of CADS.

### D.1  A TOY EXAMPLE

In this section, we visualize the effect of condition annealing on the generated samples with a simple toy example. Our goal in this experiment is to show that condition annealing does not result in sampling from unwanted regions according to the data distribution. Consider a two-dimensional Gaussian mixture model with 25 components as the data distribution, illustrated in Figure 8a. We train a conditional diffusion model with classifier-free guidance for mapping the component ID (between 1 and 25) to samples from the corresponding mixture component. In Figure 8b, we see that if we use the standard DDPM with a high guidance value, the samples roughly converge to the mode of each component. However, condition annealing on the class embeddings leads to better coverage of each component as shown in Figure 8c. Also, Figure 8c confirms that using CADS does not result in sampling from regions that are not in the support of the real data distribution.

### D.2  PERFORMANCE OF CADS VS THE NUMBER OF SAMPLING STEPS

In this section, we study the performance of CADS against the number of sampling steps, also known as neural function evaluations (NFEs). Figure 9 demonstrates that the behavior of CADS is similar to the base sampler as we change the number of sampling steps, and a consistent gap exists between sampling with and without condition annealing across different NFEs. Interestingly, with a fixed annealing schedule in CADS, the FID metric plots show a U-shaped curve over a subset of the range of NFEs in the ImageNet model. This suggests that the annealing schedule used in this experiment may be better suited for certain NFEs over others. Please note that as DDPM (Ho et al., 2020) does not perform well when using low NFEs, we switched to the DDIM sampler (Song et al., 2021a) to perform this comparison.

Table 7: Quantitative comparison among samples generated with and without CADS for various conditional generation models and diffusion samplers. CADS notably enhances the diversity of the generated samples compared to the base sampler.

| Dataset | Sampler | FID ↓ | Precision ↑ | Recall ↑ | MSS ↓ | Vendi Score ↑ |
|---|---|---|---|---|---|---|
| DeepFashion ($w_{CFG} = 4$) | DDIM | 14.60 | **0.93** | 0.02 | 0.80 | 1.03 |
| | CADS (Ours) | **7.90** | 0.76 | **0.49** | **0.35** | **2.30** |
| SHHQ ($w_{CFG} = 4$) | DDIM | 26.27 | **0.70** | 0.15 | 0.57 | 1.27 |
| | CADS (Ours) | **15.14** | 0.61 | **0.46** | **0.36** | **2.06** |
| Stable Diffusion ($w_{CFG} = 9$) | DPM++ | 45.70 | **0.70** | 0.29 | 0.19 | 5.30 |
| | CADS (Ours) | **40.35** | 0.65 | **0.42** | **0.13** | **6.93** |
| | PNDM | 45.76 | **0.68** | 0.28 | 0.19 | 5.36 |
| | CADS (Ours) | **41.37** | 0.65 | **0.38** | **0.13** | **6.83** |

### D.3 ADDITIONAL EXPERIMENTS WITH DIFFERENT SAMPLERS

Table 7 provides more comparisons on the effectiveness of CADS when used with different diffusion samplers other than DDPM (Ho et al., 2020). Specifically, we extend the results of Table 1 for the pose-to-image models using DDIM (Song et al., 2021a) as the base sampler. Additionally, we report different comparisons based on Stable Diffusion with PNDM (Liu et al., 2022b) and DPM++ (Lu et al., 2022b) methods. Similar to what has been shown in Table 3, CADS consistently improves the diversity of the base sampler without compromising quality.

### D.4 TRAINING WITH NOISY CONDITIONS

A natural extension to our approach involves training the diffusion model under noisy conditions, aiming to achieve more diverse results during inference with standard samplers. We tested this on the pose-to-image task by training the underlying diffusion process with noisy pose images. However, as indicated in Figure 10, this approach still yields outputs with limited diversity. The finding indicates that the low diversity issue lies mainly in the inference stage rather than in the training process. In fact, we found that training the diffusion model with noisy poses diminishes the effectiveness of CADS, as the network learns to associate noisy conditions with specific images.

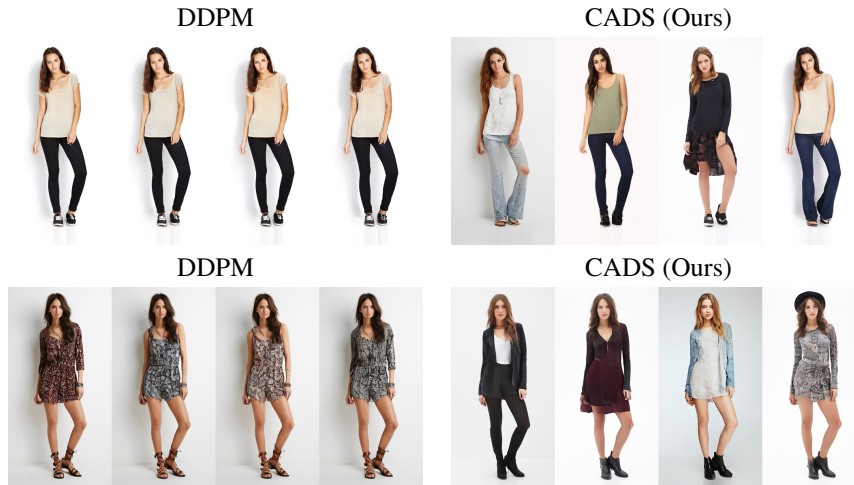

Figure 10: Comparing DDPM with CADS on a pose-to-image model trained on noisy pose images. Training on noisy images does not solve the issue of low-diversity by default, and sampling with CADS is still needed to achieve better diversity.

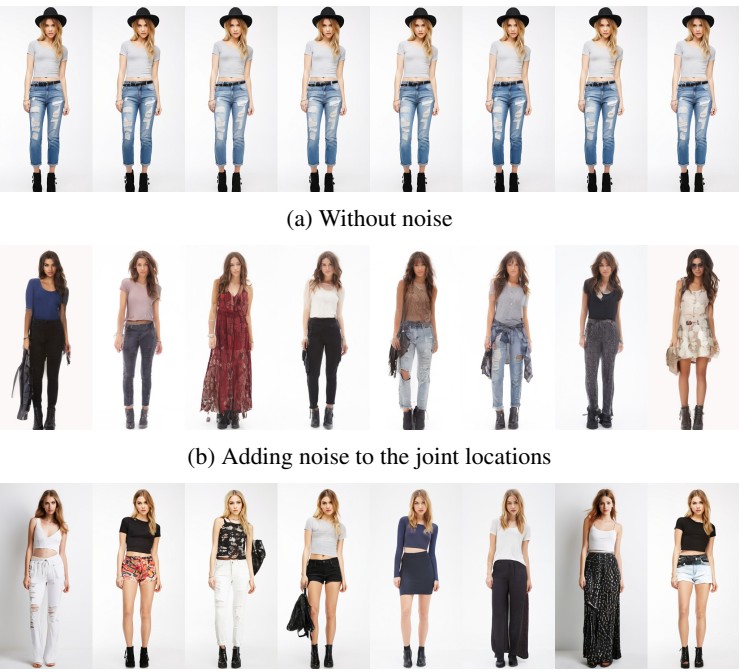

(a) Without noise

(b) Adding noise to the joint locations

(c) Adding noise to the 2D pose image

Figure 11: Comparing outputs of CADS when the noise is added to the joint locations and to the 2D pose image. Adding noise to the pose image results in better diversity and quality.

## D.5 ADDING NOISE TO THE JOINT LOCATIONS

In the pose-to-image task, we also experimented with adding noise to the joint locations instead of the entire 2D image. However, we found that this leads to blurry output images, especially at the edges between the subject and background. An illustration of this issue is given in Figure 11. Additionally, this method increases the sampling overhead since it requires the pose image to be rendered at each step. As a result, we opted to add noise directly to the pose image in subsequent experiments.

## D.6 REDUCING THE GUIDANCE SCALE

One common way to increase the diversity in diffusion models is reducing the scale of classifier-free guidance. However, similar to previous works (Dhariwal & Nichol, 2021; Ho & Salimans, 2022), we show in Figure 12 that reducing the guidance diminishes the quality of generated images. In contrast, Figure 13 shows that CADS maintains high-quality images by improving output diversity at higher guidance scales. Furthermore, when the model is trained on small datasets like DeepFashion, the low diversity issue exists at all guidance scales as evidenced by Figure 13. Therefore, reducing the guidance scale does not resolve the issue of limited diversity.

## D.7 MORE DISCUSSION ON DYNAMIC CFG

In the main text, we evaluated the performance of Dynamic CFG and CADS in the context of the class-conditional ImageNet generation model. In this section, we expand our analysis to include the pose-to-image generation task. As shown in Figure 14, both Dynamic CFG and CADS offer better diversity than DDPM; however, compared to CADS, sampling with Dynamic CFG results in more artifacts in this domain. This observation is further supported by the FID scores of 9.58 for Dynamic CFG and 7.73 for CADS. These findings reaffirm that CADS outperforms Dynamic CFG, making it a more effective solution for enhancing diversity in diffusion models.

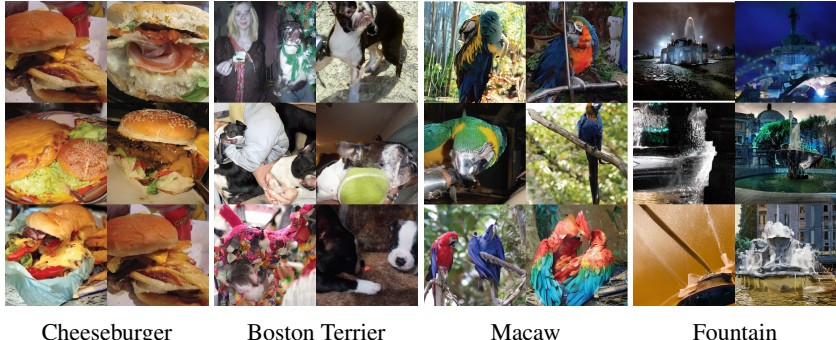

| Cheeseburger | Boston Terrier | Macaw | Fountain |

Figure 12: Examples of artifacts when using a low classifier-free guidance scale ($w_{CFG} = 1.5$) with the class-conditional model. Reducing CFG increases diversity but hurts image quality. Class labels are given below each image.

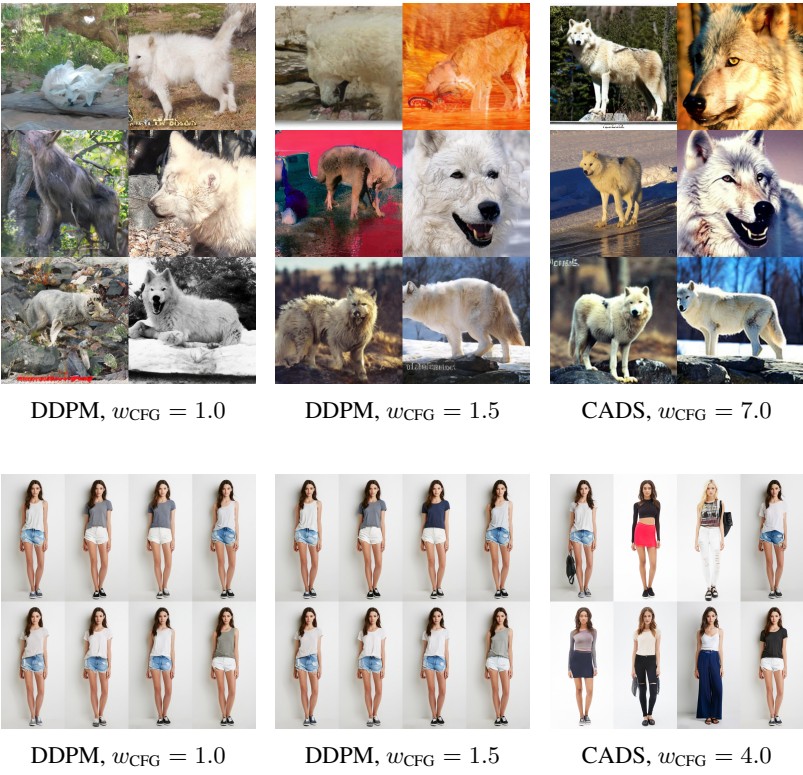

| DDPM, $w_{CFG} = 1.0$ | DDPM, $w_{CFG} = 1.5$ | CADS, $w_{CFG} = 7.0$ |

| DDPM, $w_{CFG} = 1.0$ | DDPM, $w_{CFG} = 1.5$ | CADS, $w_{CFG} = 4.0$ |

Figure 13: Examples of sampling with lower guidance values compared to high guidance values with CADS. Reducing the guidance scale deteriorates image quality in the class-conditional ImageNet model and fails to enhance diversity in the DeepFashion pose-to-image model. CADS improves diversity while preserving high-quality images in both cases.

### D.8 COMBINING DYNAMIC CFG AND CADS

We empirically evaluate the combination of Dynamic CFG and CADS for class-conditional ImageNet synthesis in Table 8 for two guidance values. Combining CADS and Dynamic CFG enhances performance at high guidance scales but degrades the FID otherwise. This suggests that combining Dynamic CFG and CADS might be beneficial only when the diversity is more limited, e.g., higher guidance scales. We chose to use CADS alone in our main experiments due to its consistent performance across all guidance values.

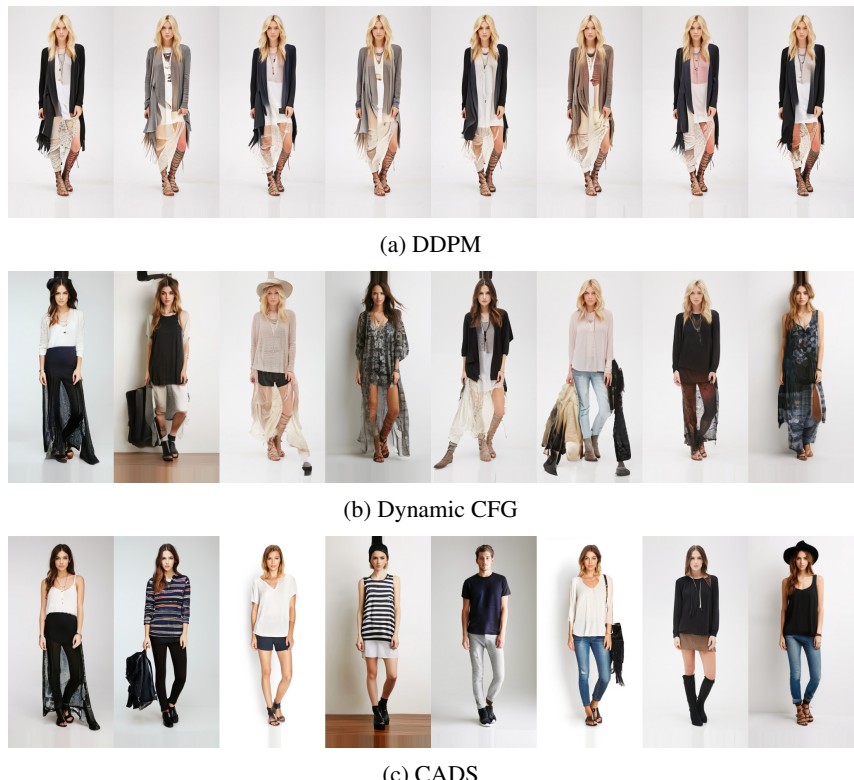

(a) DDPM

(b) Dynamic CFG

(c) CADS

Figure 14: Comparison of generated images using DDPM, Dynamic CFG, and CADS. While both Dynamic CFG and CADS have better diversity than DDPM, CADS delivers higher sample quality, as evidenced by fewer visual artifacts in each image.

Table 8: Combining Dynamic CFG with CADS can be beneficial at higher guidance scales, but it may negatively affect the metrics at lower guidance values where the diversity is less limited.

<table>
<tr><td colspan="3" align="center">(a) $w_{\mathrm{CFG}} = 5$</td><td colspan="3" align="center">(b) $w_{\mathrm{CFG}} = 2.5$</td></tr>
<tr><td>Sampling Method</td><td>FID ↓</td><td>Recall ↑</td><td>Sampling Method</td><td>FID ↓</td><td>Recall ↑</td></tr>
<tr><td>DDPM</td><td>20.83</td><td>0.32</td><td>DDPM</td><td>12.72</td><td>0.49</td></tr>
<tr><td>Dynamic CFG</td><td>18.42</td><td>0.39</td><td>Dynamic CFG</td><td>6.58</td><td>0.65</td></tr>
<tr><td>CADS</td><td>9.47</td><td>0.62</td><td>CADS</td><td>**5.02**</td><td>0.71</td></tr>
<tr><td>Dynamic CFG + CADS</td><td>**8.14**</td><td>**0.66**</td><td>Dynamic CFG + CADS</td><td>5.43</td><td>**0.74**</td></tr>
</table>

## E  ADDITIONAL ABLATIONS

In the following sections, we conduct further ablation studies to evaluate the impact of different parameters on CADS performance.

### E.1  EFFECT OF $\tau_2$

The parameter $\tau_2$ in Equation (2) determines how many steps in the beginning of sampling are run without using any information from the conditioning signal (since $\gamma(t) = 0$ for $t > \tau_2$). Table 9 illustrates the impact of varying $\tau_2$ on the class-conditional model for two guidance values as well as the pose-to-image model. Since the diversity decreases at higher guidance levels, reducing $\tau_2$ improves both FID and Recall for $w_{\mathrm{CFG}} = 5$, but negatively impacts these metrics for $w_{\mathrm{CFG}} = 2.5$. Thus, we argue that when the low diversity issue is more pronounced, more noise is needed at the beginning of inference to achieve better diversity. This phenomenon is also evident in the pose-to-image model as reducing $\tau_2$ improves the metrics there.

Table 9: Effect of changing $\tau_2$ on the class-conditional model for two guidance values as well as the pose-to-image model. We use a fixed $\tau_1 = 0.5$ for each table.

(a) $w_{\text{CFG}} = 5$

| $\tau_2$ | FID $\downarrow$ | Recall $\uparrow$ |
|------|------|------|
| 0.5 | **8.21** | **0.67** |
| 0.9 | 9.47 | 0.62 |

(b) $w_{\text{CFG}} = 2.5$

| $\tau_2$ | FID $\downarrow$ | Recall $\uparrow$ |
|------|------|------|
| 0.5 | 6.14 | **0.76** |
| 0.9 | **5.02** | 0.71 |

(c) Pose-to-image model

| $\tau_2$ | FID $\downarrow$ | Recall $\uparrow$ |
|------|------|------|
| 0.5 | **8.26** | **0.38** |
| 1 | 10.93 | 0.19 |

Table 10: Comparing different choices of $\gamma(t)$ based on a polynomial annealing schedule with degree $d$ and fixed $\tau = 0.6$. The piece-wise linear function for $\gamma(t)$ with $\tau_1 = 0.6$ and $\tau_2 = 0.8$ outperforms all polynomial settings. We use $s = 0.15$ for all entries.

| $\gamma(t)$ | FID $\downarrow$ | Recall $\uparrow$ |
|------|------|------|
| Baseline | **14.20** | **0.51** |
| Polynomial, $d = 1$ | 15.85 | 0.45 |
| Polynomial, $d = 2$ | 14.98 | 0.48 |
| Polynomial, $d = 3$ | 14.49 | 0.48 |
| Polynomial, $d = 4$ | 14.32 | 0.50 |

## E.2 VARYING THE ANNEALING SCHEDULE

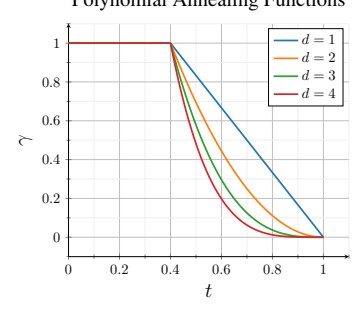

Polynomial Annealing Functions

In this section, we investigate the impact of different $\gamma(t)$ functions on the performance of CADS. Specifically, we employ polynomial functions of varying degrees for $\gamma(t)$ and compare them with a piecewise linear function. The polynomial annealing schedule is formulated as

$$\gamma(t) = \begin{cases} 1 & t \leq \tau, \\ \left(\frac{1-t}{1-\tau}\right)^d & t > \tau. \end{cases} \quad (17)$$

Table 10 indicates that FID and Recall only slightly change as we increase the degree of the polynomial, and the piecewise linear function with appropriate $\tau_1$ and $\tau_2$ offers better scores. Therefore, we opted for the piecewise linear annealing function due to its better performance and interpretability.

## E.3 CHANGING DISTRIBUTION OF THE NOISE

We now examine various noise distributions, namely Uniform, Gaussian, Laplace, and Gamma, for the noise parameter $\boldsymbol{n}$ introduced in Equation (1). Table 11 illustrates that CADS is not sensitive to the particular distribution of the added noise. The only influential factor is the standard deviation of $\boldsymbol{n}$, which can be absorbed into the noise scale $s$ as defined in Equation (1). Based on this, we used Gaussian noise in all of our experiments.

## F LIMITATION OF IS AND PRECISION

This section demonstrates that IS and Precision have limitations in accurately evaluating diverse inputs, suggesting cautious interpretation of these metrics. To validate this, we compute the metrics for 10k randomly sampled images from the ImageNet training and evaluation set and report the results in Table 12. As anticipated, FID and Recall show superior performance on the real data. In contrast, DDPM samples exhibit much better Precision and IS scores, largely due to their limited diversity. When sampling with CADS, IS and Precision are lower than DDPM but they do not fall bellow the values computed from the real data. This indicates that CADS enhances diversity while maintaining high image quality.

Table 11: Comparing different noise distributions for $n$ in Equation (1) on class-conditional ImageNet generation with $w_{\mathrm{CFG}} = 5$. All samples for this table are generated using CADS with $\tau_1 = 0.5$, $\tau_2 = 0.9$, $s = 0.15$, and $\psi = 1.0$. Gamma distributions are centered to have zero mean.

| Noise Distribution | FID $\downarrow$ | Recall $\uparrow$ |
|---|---|---|
| $\mathrm{Normal}(0,1)$ | 9.47 | 0.62 |
| $\mathrm{Normal}(0,2)$ | 8.89 | 0.64 |
| $\mathrm{Laplace}(0,1)$ | 9.15 | 0.63 |
| $\mathrm{Laplace}(0,\sqrt{1.5})$ | 8.86 | 0.63 |
| $\mathrm{Gamma}(1,1)$ | 9.38 | 0.62 |
| $\mathrm{Gamma}(4,4)$ | 8.78 | 0.63 |
| $\mathrm{Gamma}(4,\sqrt{4})$ | 9.68 | 0.60 |
| $\mathrm{Gamma}(7,\sqrt{7})$ | 9.67 | 0.61 |
| $\mathrm{Uniform}(0,1)$ | 13.42 | 0.50 |
| $\mathrm{Uniform}(-1,1)$ | 10.54 | 0.59 |
| $\mathrm{Uniform}(-3,3)$ | 8.89 | 0.63 |

Table 12: Comparing metrics evaluated based on generated and real images. IS and Precision show artificially high values for DDPM, suggesting that FID captures realism and diversity better than these two metrics.

| Samples | FID $\downarrow$ | IS $\uparrow$ | Precision $\uparrow$ | Recall $\uparrow$ |
|---|---|---|---|---|
| real, train | **4.12** | 332.49 | 0.75 | **0.76** |
| real, eval | **4.86** | 235.85 | 0.75 | **0.74** |
| DDPM | 20.83 | **488.42** | **0.92** | 0.32 |
| CADS | 9.47 | 417.41 | 0.82 | 0.62 |

## G  IMPLEMENTATION DETAILS

**Sampling algorithms**   We provide the pseudocode for the annealing schedule and noise addition in Figure 15. In addition, the detailed algorithms for sampling with CADS and Dynamic CFG are provided in Algorithms 1 and 2.

**Sampling and evaluation details**   Table 13 includes the sampling hyperparameters used to create each table in the main text. For computational efficiency, 10k samples are employed to calculate the FID across in all the experiments, except the state-of-the-art results (Table 2), where 50k samples are used to align with the current literature. We use 250 sampling steps for computing the metrics for Table 2 and 100 sampling steps for all other experiments. For evaluating the Stable Diffusion model, we sample 1000 random captions from the COCO validation set (Lin et al., 2014) and generate 10 samples per prompt.

**Training details for the DeepFashion and SHHQ models**   We adopt the same network architectures and hyperparameters as those in the publicly released LDM models (Rombach et al., 2022), available at `https://github.com/CompVis/latent-diffusion`. Specifically, we train a VQGAN autoencoder (Esser et al., 2021) on the DeepFashion and SHHQ datasets from scratch with the adversarial and reconstruction losses. Then, we fit a diffusion model in the VQGAN's latent space using the velocity prediction formulation (Salimans & Ho, 2022). For detailed information on the exact hyperparameters of the models employed, please refer to the official codebase of LDM, along with the original papers (Esser et al., 2021; Rombach et al., 2022).

## H  MORE VISUAL RESULTS

This section provides more qualitative results to demonstrate the effectiveness of CADS compared to DDPM. Figures 16 to 18 show more samples from the class-conditional ImageNet generation task at both 256×256 and 512×512 resolutions. We also provide more samples from the pose-

Table 13: Sampling parameters used in different experiments.

| | Dataset | $w_{\text{CFG}}$ | $\tau_1$ | $\tau_2$ | $s$ | $\psi$ |
|---|---|---|---|---|---|---|
| Parameters for Table 1 | DeepFashion | 4 | 0.35 | 0.7 | 1 | 0 |
| | SHHQ | 4 | 0.425 | 0.6 | 0.8 | 0 |
| | ImageNet 256 | 5 | 0.5 | 0.9 | 0.15 | 1 |
| | ImageNet 512 | 5 | 0.6 | 1 | 0.1 | 1 |
| | ID3PM | 4 | 0.5 | 1 | 0.5 | 1 |
| | Stable Diffusion | 9 | 0.6 | 0.9 | 0.25 | 1 |
| Parameters for Table 2 | ImageNet 256 | 2 | 0.525 | 0.9 | 0.06 | 1 |
| | | 2.25 | 0.5 | 0.9 | 0.06 | 1 |
| | | 2.5 | 0.475 | 0.9 | 0.06 | 1 |
| | ImageNet 512 | 2 | 0.625 | 1.0 | 0.07 | 1 |
| | | 2.25 | 0.6 | 1.0 | 0.07 | 1 |
| | | 2.5 | 0.575 | 1.0 | 0.07 | 1 |
| Parameters for Figure 5 | ImageNet 256 | 1 | 0.8 | 0.9 | 0.05 | 1 |
| | | 1.25 | 0.75 | 0.9 | 0.05 | 1 |
| | | 1.5 | 0.7 | 0.9 | 0.05 | 1 |
| | | 2 | 0.55 | 0.9 | 0.075 | 1 |
| | | 2.5 | 0.525 | 0.9 | 0.075 | 1 |
| | | 3 | 0.5 | 0.9 | 0.1 | 1 |
| | | 4 | 0.5 | 0.9 | 0.1 | 1 |
| | | 5 | 0.5 | 0.9 | 0.15 | 1 |
| | | 7 | 0.475 | 0.9 | 0.2 | 1 |
| Parameters for Table 6a | ImageNet 256 | 5 | 0.6 | 1 | - | 0 |
| Parameters for Table 6b | ImageNet 256 | 5 | - | 1 | 0.1 | 0 |
| Parameters for Table 6c | ImageNet 256 | 5 | 0.6 | 1 | 0.1 | - |

to-image model in Figures 19 to 21. Finally, Figures 22 and 23 contain more samples from the identity-conditioned face synthesis network.

```
1  def linear_schedule(t, tau1, tau2):
2    if t <= tau1:
3        return 1.0
4    if t >= tau2:
5        return 0.0
6    gamma = (tau2 - t)/(tau2 - tau1)
7    return gamma
8
9  def add_noise(y, gamma, noise_scale, psi, rescale=False):
10   y_mean, y_std = mean(y), std(y)
11   y = sqrt(gamma) * y + noise_scale * sqrt(1 - gamma) * randn_like(y)
12   if rescale:
13       y_scaled = (y - mean(y)) / std(y) * y_std + y_mean
14       y = psi * y_scaled + (1 - psi) * y
15   return y
16
```

Figure 15: Pseudocode for the annealing schedule function and adding noise to the condition.

---

**Algorithm 1** Sampling with CADS

**Require:** $w_{CFG}$: classifier-free guidance strength
**Require:** $\boldsymbol{y}$: Input condition
**Require:** Annealing schedule function $\gamma(t)$, initial noise scale $s$
1: Initial value: $\boldsymbol{z}_1 \sim \mathcal{N}(\boldsymbol{0}, \boldsymbol{I})$
2: **for** $t = T, \ldots, 1$ **do**

  ○ Add noise to the input condition and the null condition
3:  $\hat{\boldsymbol{y}} = \sqrt{\gamma(t)}\boldsymbol{y} + s\sqrt{1 - \gamma(t)}\boldsymbol{n}, \quad \hat{\boldsymbol{y}}_{\text{null}} = \sqrt{\gamma(t)}\boldsymbol{y}_{\text{null}} + s\sqrt{1 - \gamma(t)}\boldsymbol{n}'$
4:  Rescale $\hat{\boldsymbol{y}}$ and $\hat{\boldsymbol{y}}_{\text{null}}$ if needed

  ○ Compute the classifier-free guided output at $t$
5:  $\hat{D}_{\text{CFG}}(\boldsymbol{z}_t, t, \boldsymbol{y}) = D(\boldsymbol{z}_t, t, \hat{\boldsymbol{y}}_{\text{null}}) + w_{\text{CFG}}(D(\boldsymbol{z}_t, t, \hat{\boldsymbol{y}}) - D(\boldsymbol{z}_t, t, \hat{\boldsymbol{y}}_{\text{null}}))$

  ○ Perform one sampling step (e.g. one step of DDPM)
6:  $\boldsymbol{z}_{t-1} = \texttt{diffusion\_reverse}(\hat{D}_{\text{CFG}}, \boldsymbol{z}_t, t)$
7: **end for**
8: **return** $\boldsymbol{z_0}$

---

**Algorithm 2** Sampling with Dynamic CFG

**Require:** $w_{CFG}$: classifier-free guidance strength
**Require:** $\boldsymbol{y}$: Input condition
**Require:** Schedule function $\gamma(t)$
1: Initial value: $\boldsymbol{z}_1 \sim \mathcal{N}(\boldsymbol{0}, \boldsymbol{I})$
2: **for** $t = T, \ldots, 1$ **do**

  ○ Modulate the guidance weight according to the time step
3:  $\hat{w}_{\text{CFG}} = \gamma(t)w_{\text{CFG}}$

  ○ Compute the classifier-free guided output at $t$
4:  $\hat{D}_{\text{CFG}}(\boldsymbol{z}_t, t, \boldsymbol{y}) = D(\boldsymbol{z}_t, t, \boldsymbol{y}_{\text{null}}) + \hat{w}_{\text{CFG}}(D(\boldsymbol{z}_t, t, \boldsymbol{y}) - D(\boldsymbol{z}_t, t, \boldsymbol{y}_{\text{null}}))$

  ○ Perform one sampling step (e.g. one step of DDPM)
5:  $\boldsymbol{z}_{t-1} = \texttt{diffusion\_reverse}(\hat{D}_{\text{CFG}}, \boldsymbol{z}_t, t)$
6: **end for**
7: **return** $\boldsymbol{z_0}$

DDPM                                                    CADS (Ours)

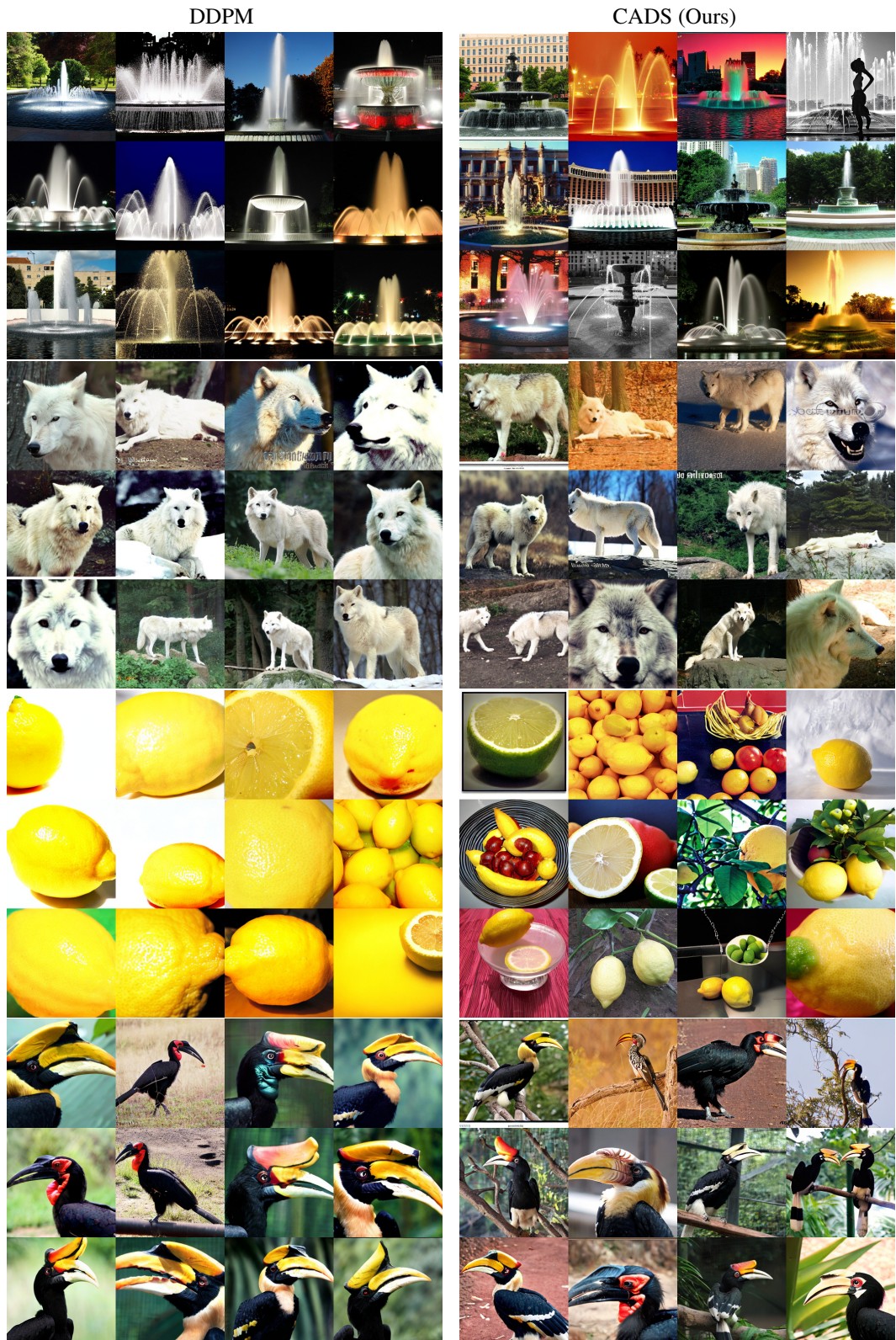

Figure 16: Uncurated examples from the class-conditional ImageNet $256 \times 256$ model with $w_{\text{CFG}} = 5$. Class labels from top to bottom: Fountain (562), Alaskan Wolf (270), Lemon (951), and Hornbill (93).

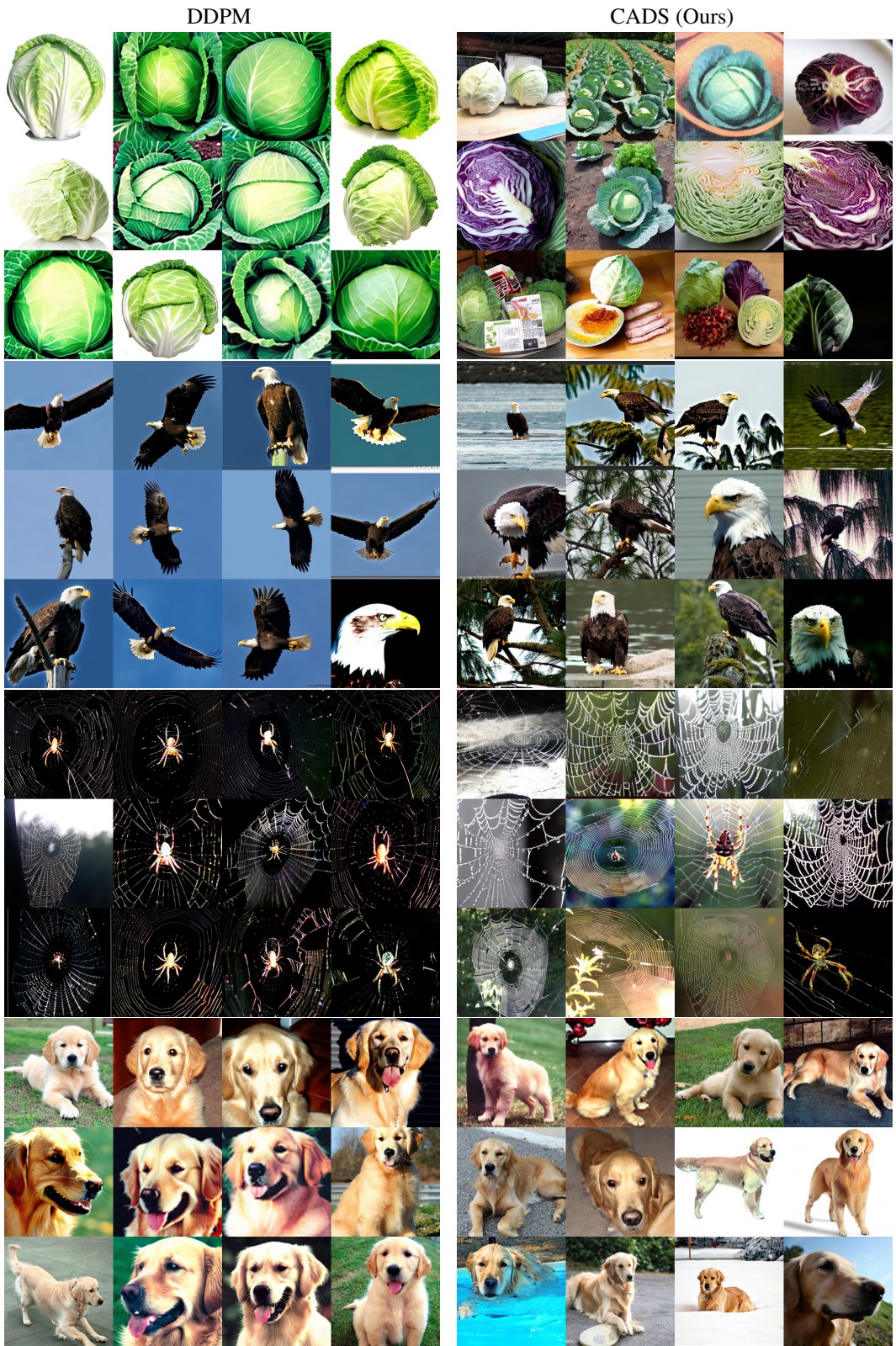

Figure 17: Uncurated examples from the class-conditional ImageNet 256×256 model with $w_{\text{CFG}} = 7$. Class labels from top to bottom: Cabbage (936), Bald Eagle (22), Spider's Web (815), and Golden Retriever (207).

DDPM CADS (Ours)

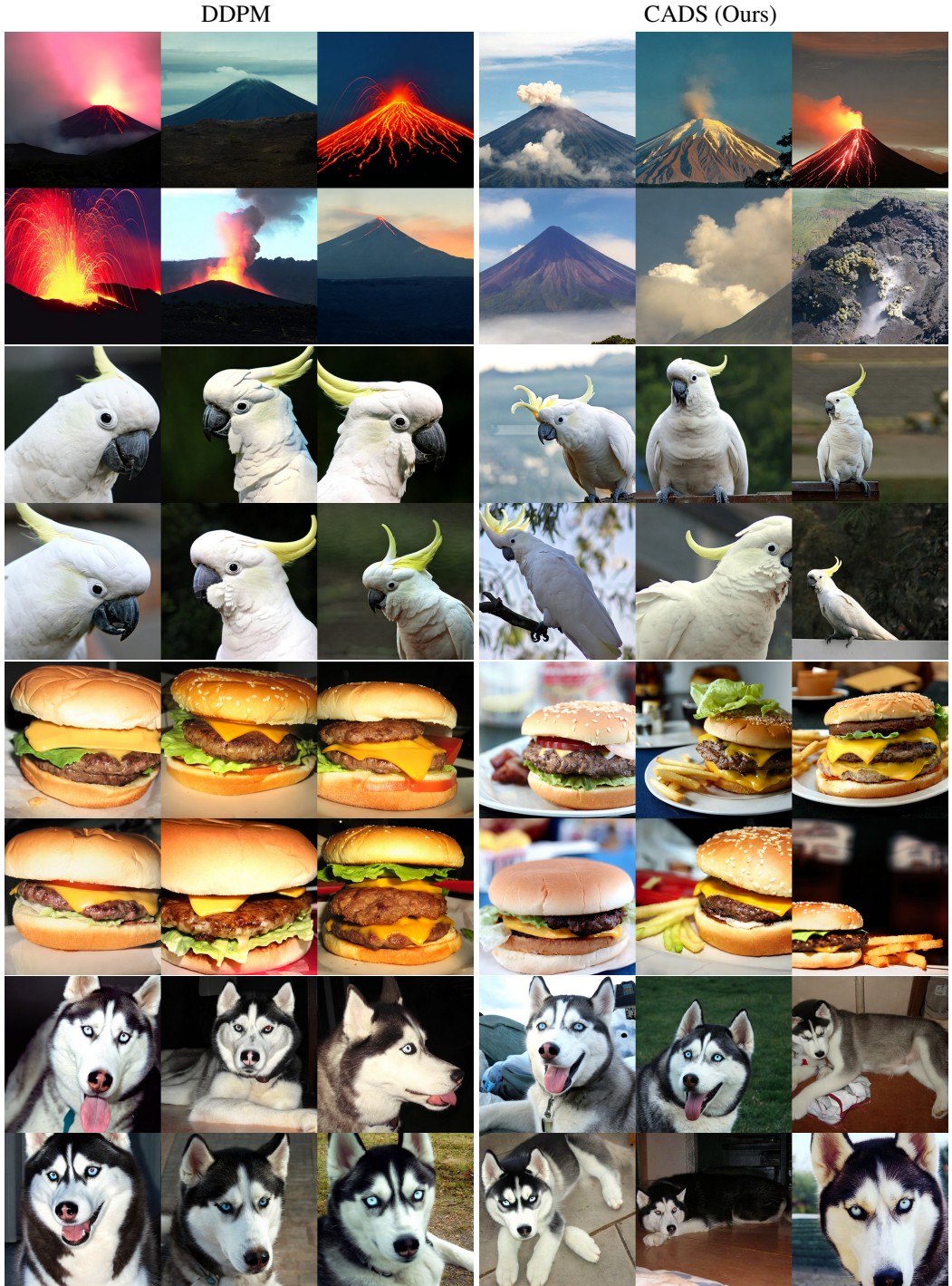

Figure 18: Uncurated samples from the class-conditional ImageNet $512\times512$ model with $w_{\mathrm{CFG}} = 6$. Class labels from top to bottom are: Volcano (980), Sulphur-Crested Cockatoo (89), Cheeseburger (933), and Siberian Husky (250).

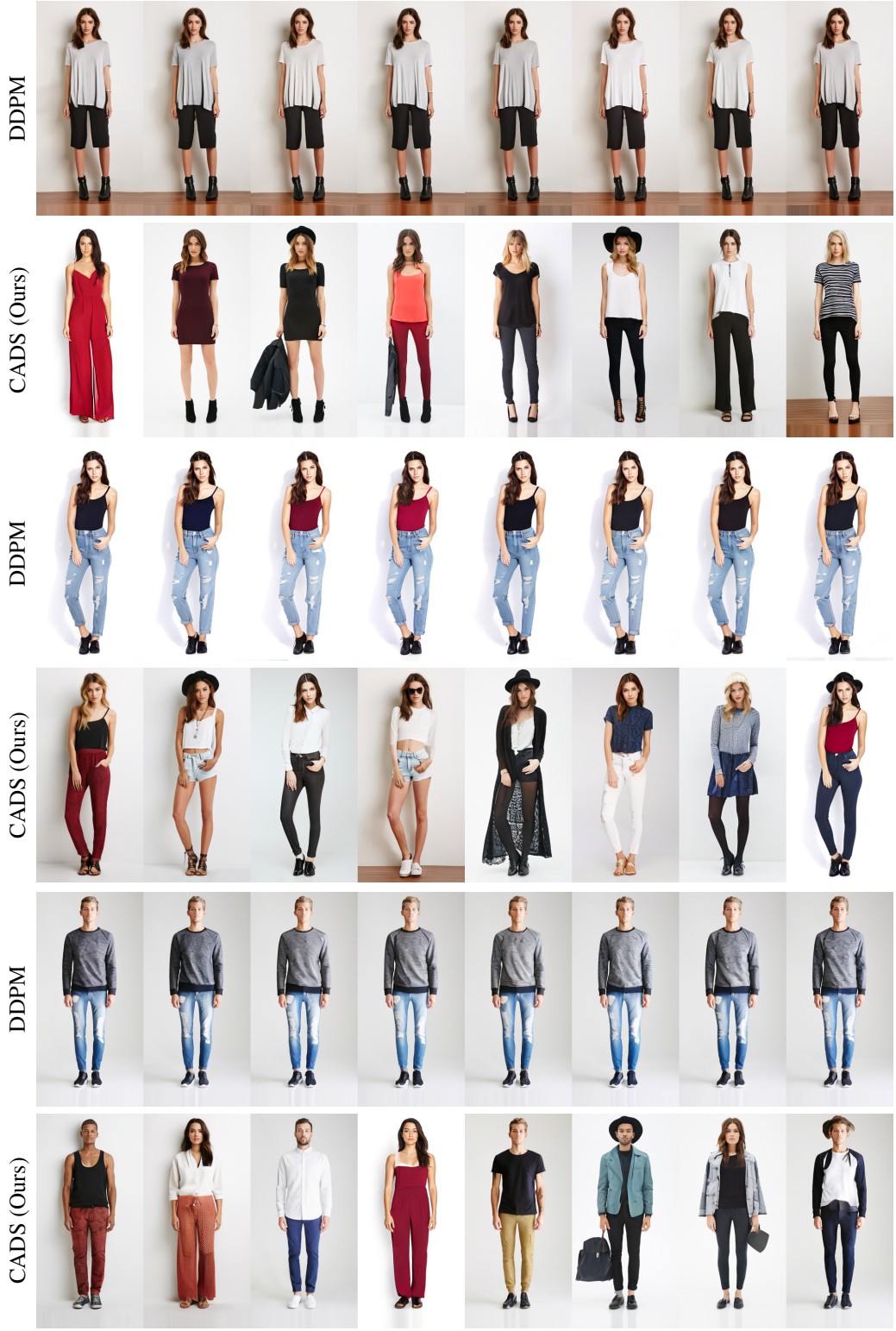

Figure 19: More samples from the DeepFashion pose-to-image model. Each row represents generated images from a fixed pose with 8 different seeds. Random seeds are identical between samples of DDPM and CADS.

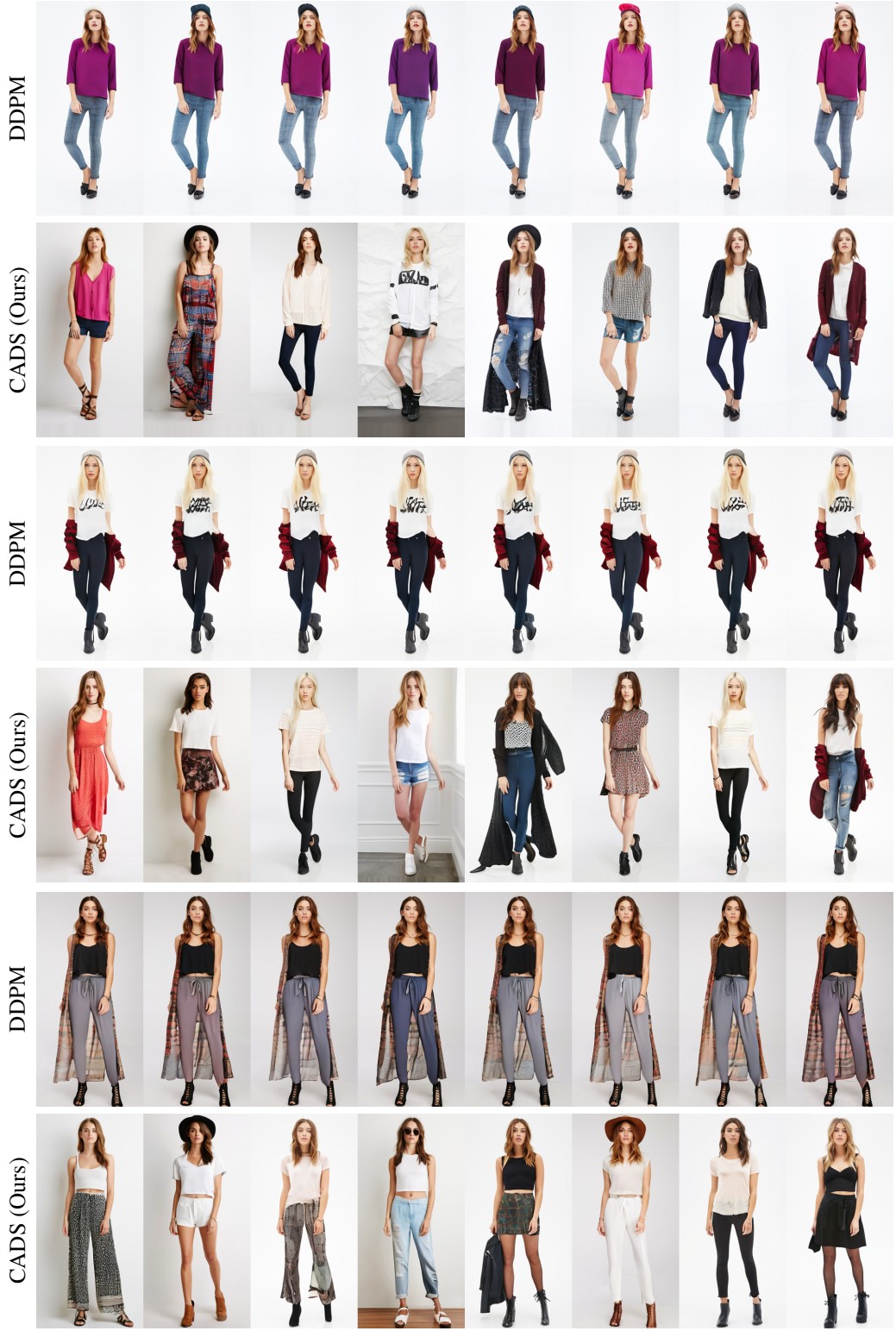

Figure 20: More samples from the DeepFashion pose-to-image model. Each row represents generated images from a fixed pose with 8 different seeds. Random seeds are identical between samples of DDPM and CADS.

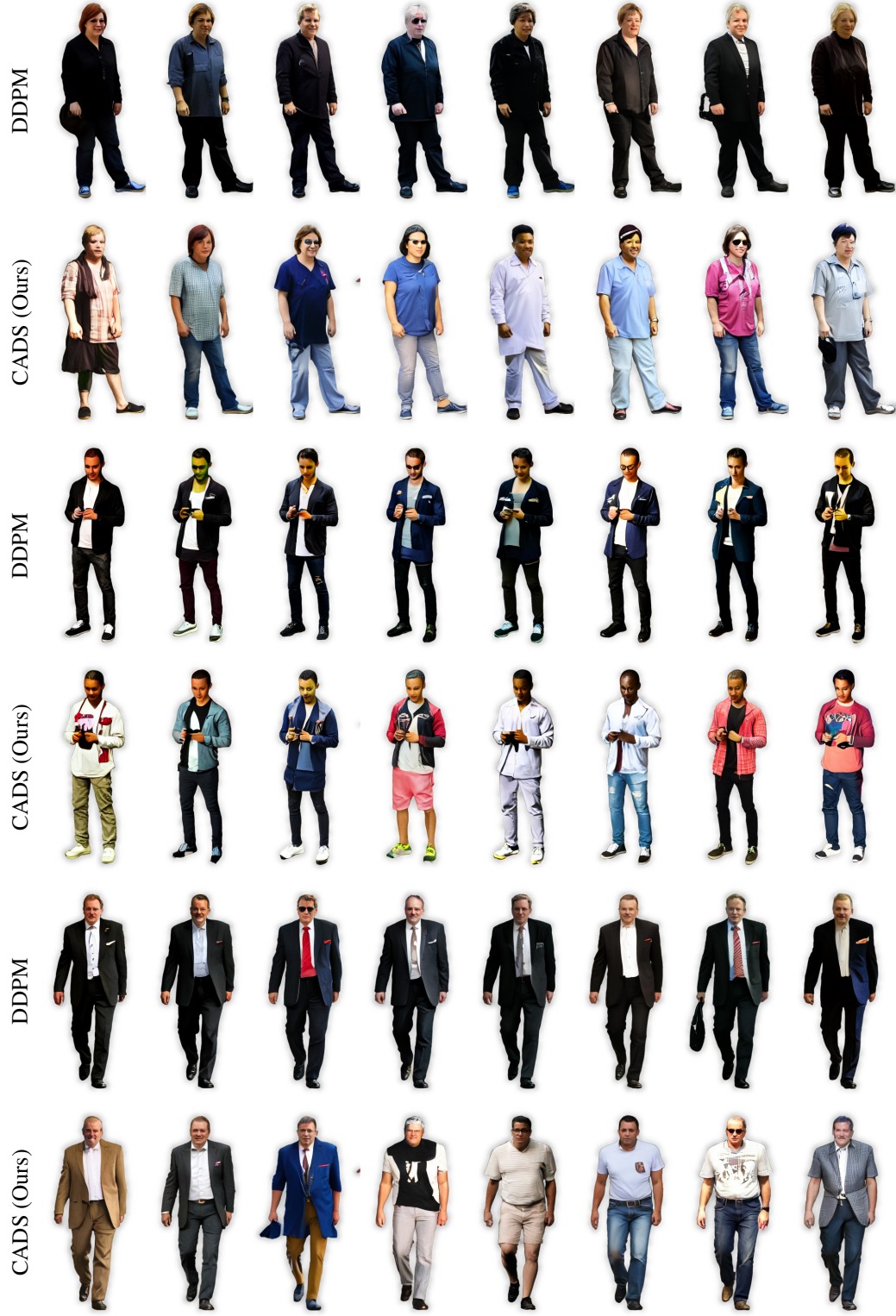

Figure 21: More samples from the SHHQ pose-to-image model. Each row represents generated images from a fixed pose with 8 different seeds. Random seeds are identical between samples of DDPM and CADS.

DDPM                                    CADS (Ours)

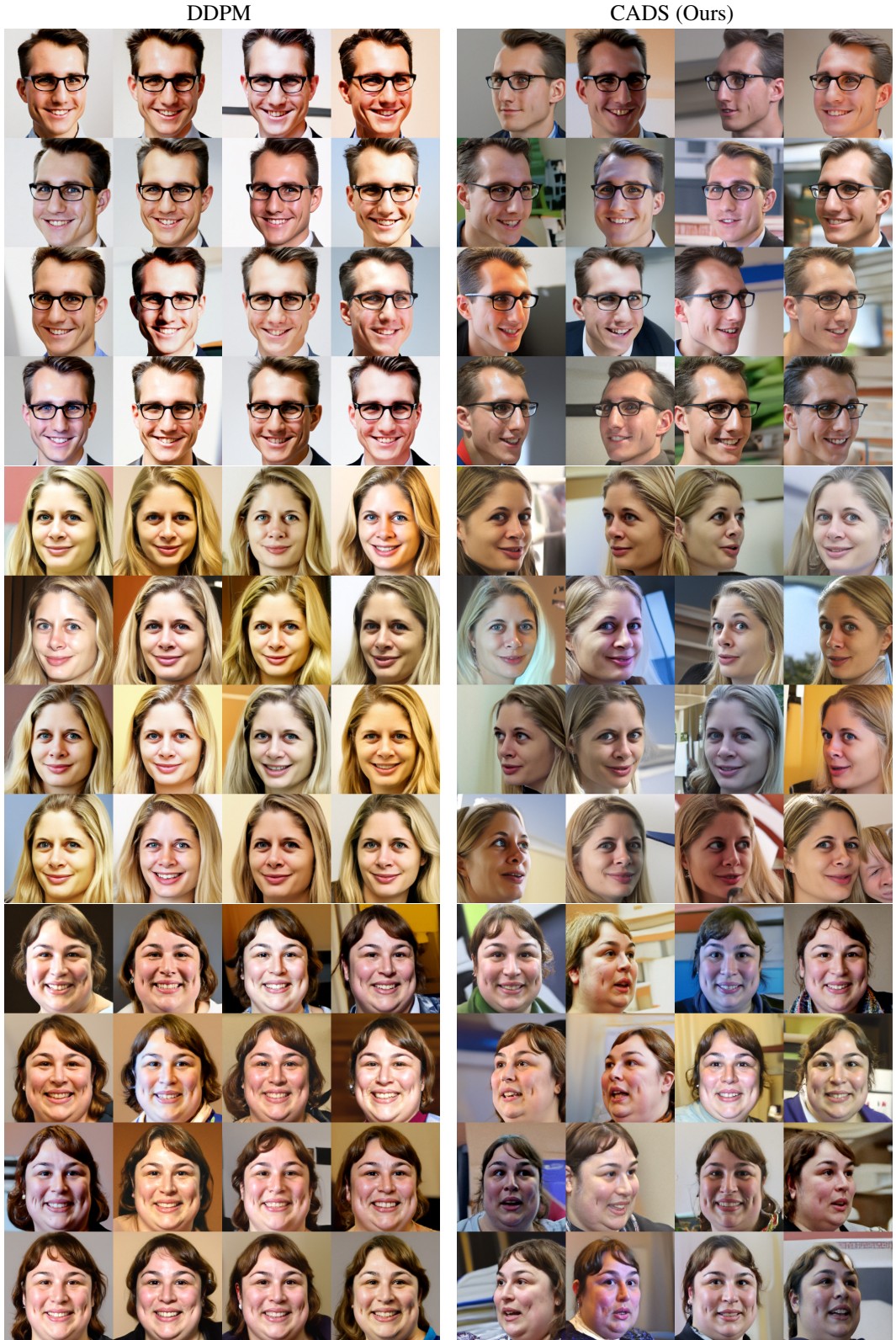

Figure 22: More results from the identity-conditioned face generation task with ID3PM. Each 4×4 grid contains generated images using a fixed identity vector as the input.

DDPM           CADS (Ours)

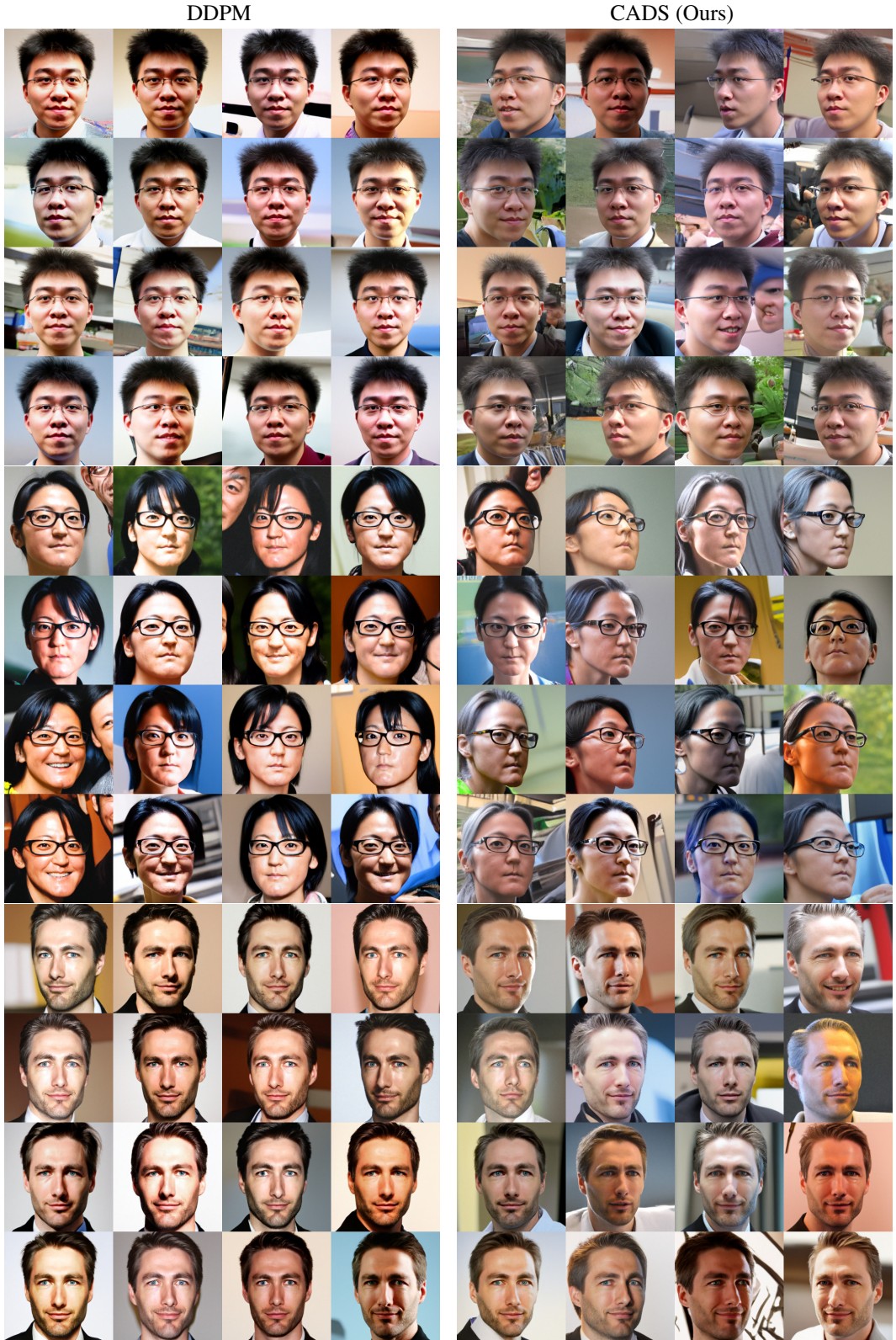

Figure 23: More results from the identity-conditioned face generation task with ID3PM. Each 4×4 grid contains generated images using a fixed identity vector as the input.

