# OpenReview forum: "CADS: Unleashing the Diversity of Diffusion Models through Condition-Annealed Sampling"
_ICLR.cc/2024/Conference — ICLR 2024 spotlight_

### Official Review · Reviewer_Djqj · 2023-10-30

**Soundness:** 3 good
**Presentation:** 3 good
**Contribution:** 3 good
**Rating:** 8
**Confidence:** 4

**Summary:**

In this paper, conditional-annealed diffusion sampler (CADS) is proposed to address the limited diversity of diffusion models with a high classifier-free guidance scale or when trained on small datasets. Specifically, during early inference, the conditional signal is perturbed largely and then restored gradually  during late inference. The proposed method is simple and effective, amplifying the diversity of the generated samples. Besides, it requires minimal computational overhead and easy to implement. In addition, this paper provides extensive experiments on various tasks and ablation studies, validating the effectiveness and novelty of the proposed method.

**Strengths:**

+ The proposed method is simple but effective. It is easy to implement and validated with various experiments. Besides, it outperforms a naive and intuitive approach, termed as adaptive CFG, confirming its technical contributions.
+ To validate the proposed method, this paper provides extensive experiments on various tasks and ablation studies. Thus, it is a solid paper.

**Weaknesses:**

- In Appendix G, this paper provides sampling and evaluation details, shown in Table 12. The sampling hyperparameters is set deliberately for these experiments. To my knowledge, the training setting is mostly fixed for the same dataset even though it is an ablation study. But, as shown in Table 12, this rule is broken. For example, the setting of ImageNet 256 in Table 1 is different from the setting of ImageNet 256 in Table 6a. How do you set the sampling hyperparameters?
- As shown in Table 2, there is a missing comparison. The authors should provide the result of DiT-XL/2 with CADS($w_{CFG=1.5}$) or the result of DiT-XL/2 ($w_{CFG=2}$), alleviating the effects of different cfgs.

**Questions:**

- Empirically, with high guidance weights, the diffusion sampler can produce unnatural images and sometimes even diverges. There is a concern about the proposed method: is it easy to produce bad images?

---

> ### Author Response · Authors · 2023-11-16
> **Official Response to Reviewer Djqj**
>
> ### **Ablations and hyperparameter selection**
>
> The reviewer’s question brings up an interesting point about the hyperparameter settings in the present work.  The settings described in our paper were determined heuristically based on experimentation with the individual datasets.  It is important to note that, as our method is a *sampling* method, no hyperparameter choices need to be made at training time, and experimentation to help select hyperparameter settings is relatively inexpensive.  We also found that our method is quite robust to a range of settings.  When performing an ablation study on individual hyperparameters, the other hyperparameters must be kept fixed at predetermined values.  These values were selected to best illustrate the effects of exploring the range of the *ablated* hyperparameters, which in some cases differ from the settings used elsewhere in the paper.  However, we agree with the reviewer about the importance of maintaining a direct comparison between experiments, and we report below additional ablations with the fixed hyperparameters chosen to match those in Table 1.  These results comport with the conclusions of the original ablation study while, we hope, better connecting with the earlier experiments.
>
> **New ablation on $\tau_1$**
>
> | **$\tau_1$** | **FID** | **Recall** |
> | :---: | :---: | :---: |
> | 0.2 | 47.21 | 0.81 |
> | 0.5 | 9.47 | 0.62 |
> | 0.9 | 20.64 | 0.33 |
>
> **New ablation on $\psi$**
>
> | $\psi$ | FID | Recall |
> | :---: | :---: | :---: |
> | 0.0 | 111.39 | 0.81 |
> | 0.5 | 11.55 | 0.60 |
> | 1.0 | 9.47 | 0.62 |
>
>
> **New ablation on $s$ with and without rescaling**
>
> | $s$ | FID ($\psi=0$) | Recall ($\psi=0$) | FID ($\psi=1$) | Recall ($\psi=1$) |
> | :---: | :---: | :---: | :---: | :---: |
> | 0.025 | 16.88 | 0.43 | 12.94 | 0.50 |
> | 0.05 | 14.96 | 0.48 | 12.69 | 0.52 |
> | 0.10 | 14.13 | 0.63 | 10.18 | 0.60 |
> | 0.15 | 111.39 | 0.81 | 9.47 | 0.62 |
>
> ### **Missing comparisons in Table 2**
>
> We thank the reviewer for pointing out the missing comparisons in Table 2. Please find the completed version below. We have added the metrics for regular CFG at guidance levels that produced best FID at both resolutions. This further confirms the fact that CADS allows the usage of higher guidance values without a significant drop in diversity/FID.
>
> | **ImageNet 256**|  |  |  |  |
> | --- | :---: | :---: | :---: | :---: |
> | **Sampler** | **FID** | **IS** | **Precision** | **Recall** |
> | DDPM ($w_{cfg}=1.5$) | 2.27 | 278.24 | 0.83 | 0.57 |
> | DDPM ($w_{cfg}=2.0$) | 5.82 | 379.39 | 0.89 | 0.48 |
> | CADS ($w_{cfg}=2.0$) | 1.70 | 268.77 | 0.78 | 0.64 |
>
>
> | **ImageNet 512**|  |  |  |  |
> | --- | :---: | :---: | :---: | :---: |
> | **Sampler** | **FID** | **IS** | **Precision** | **Recall** |
> | DDPM ($w_{cfg}=1.5$) | 3.04 | 240.82 | 0.84 | 0.54 |
> | DDPM ($w_{cfg}=2.5$) | 10.20 | 388.81 | 0.85 | 0.35 |
> | CADS ($w_{cfg}=2.5$) | 2.31 | 239.56 | 0.80 | 0.61 |
>
> ### **High guidance values and producing bad images**
>
> The short answer is that, within a wide range of settings, it is quite difficult to produce bad images using our method.  None of the images in our paper were cherry-picked or otherwise curated.  However, it is important to note that CADS was developed to solve the issue of limited image diversity when using high guidance scales and was not designed to protect against the deleterious effects of ultra-high guidance scales.  As our method is built on top of CFG, we suspect that it is similarly susceptible to these effects.  However, our method provides practitioners with a simple tool for avoiding the diversity limitations of CFG, which we argue increases the range of scales that can be used in practice versus standard methods.

---

> > ### Comment · Reviewer_Djqj · 2023-11-22
> >
> > I have read authors’ rebuttal. The authors solve my concerns. Thus, I prefer to raise my score to 8.

---

> > > ### Author Response · Authors · 2023-11-22
> > > **Official Response to Reviewer Djqj**
> > >
> > > We would like to thank the reviewer once again for providing helpful comments and for supporting our work.

---

> ### Comment · Area_Chair_94LA · 2023-11-21
>
> Reviewer Djqj,
>
> As the authors' rebuttal had been submitted, does it address your concerns?
>
> Please reply to the rebuttal, and pose the final decision as well.
>
> Thanks,
> AC

---

### Official Review · Reviewer_iK6t · 2023-11-02

**Soundness:** 4 excellent
**Presentation:** 3 good
**Contribution:** 3 good
**Rating:** 8
**Confidence:** 4

**Summary:**

Although conditional Diffusion Models have shown impressive performance in good coverage of the real data distribution, it is still limited in covering all the modes of the complex real distribution. In this paper, the authors are proposing a simple but effective sampling method of pretrained Diffusion Models for sampling more diverse results without additional time cost. Comprehensive experiments demonstrate the performance improvements of the proposed method.

**Strengths:**

- Good paper writing. Most parts of the paper is written understandable and reasonable.
- Simple but effective method.
- Performance improvement (w.r.t. diversity) in Pose-to-image generation is impressive.
- Experiments are done comprehensively.
- The proposed method is theoretically analyzed and also proven to be effective by toy dataset experiment. (in Appendix)

**Weaknesses:**

1. “Adaptive” may not be an appropriate term. To my knowledge, ‘adaptive’ is used for a method of which parameters are dynamically changed depending on a given input value [1]. Here, $z_t$ is the input I was expecting rather than $t$ since $t$ is a value within a fixed time window.

2. There are a lot of sampling methods including DDIM [2] while only the original DDPM sampler is used to compare. Considering the fact that DDIM is a more widely used sampling method than the original DDPM sampling method, additional comparison with DDIM is needed.


[1] Arbitrary Style Transfer in Real-time with Adaptive Instance Normalization, ICCV’17
[2] Denoising Diffusion Implicit Models, ICLR’21

**Questions:**

1. The definition of “diversity” is provided right under Section 3.
What is the meaning of “random seed”? Does it mean the seed of the randomness (e.g., random.seed(1) in Python) or different $x_T$? To me, comparing samples from different seed is less clear than comparing different samples from the fixed seed.
For example, we have two $x_T$ sampled from the standard normal, i.e., $x_T^{(0)}$, $x_T^{(1)}$. Let’s say the regular sampling method is $f$ (e.g., DDPM), the proposed sampling method is $g$, and an arbitrary metric of the semantic distance between two images is $h$.
To me, the effect of the proposed method would be understandable if $h(f(x_T^{(0)}), f(x_T^{(1)})) < h(g(x_T^{(0)}), g(x_T^{(1)}))$ because we can consider that more modes in the real distribution are covered by the standard gaussian. However, if the noise samples $x_T^{(0)}$, $x_T^{(1)}$ are sampled from different random seed respectively, maybe it's a trivial issue, but it sounds somehow unclear to me. Further clarifications on this point are needed.

2. What is the justification for the higher IS of DDPM across most of $w_{\text{CFG}}$ in Fig. 5?

---

> ### Author Response · Authors · 2023-11-16
> **Official Response to Reviewer iK6t**
>
> ### **Adaptive CFG terminology**
>
> We thank the reviewer for pointing this out.  We have changed the terminology to *dynamic* CFG to avoid confusion with adaptive normalization scenarios that are data dependent.
>
> ### **More experiments with DDIM and other samplers**
>
> We appreciate the reviewer’s point regarding the samplers one has to choose from.  Table 3 in our submission reports our experiments with other samplers based on the class-conditional ImageNet model. In our revised manuscript, we have added a section in the appendix that extends the experiments reported in Table 1 with additional samplers. Specifically, we compared our results on DDIM for both pose-to-image models and reported additional experiments using DPM++ [1] and PNDM [2] samplers with StableDiffusion. The table reporting these additional experiments is shown below.  In addition, our response to Reviewer fxon also includes an experiment of performance-versus-NFE based on DPM++ and DDIM samplers. We observed that across all tested scenarios and samplers, CADS consistently improves the diversity of output over that of the base sampler while maintaining image quality.
>
> | **Dataset** | **Sampler** | **FID** ↓ | **Precision** ↑ | **Recall** ↑ | **MSS** ↓ | **Vendi Score** ↑ |
> |:---------|:---------|:---------:|:---------:|:---------:|:---------:|:---------:|
> | DeepFashion | DDIM | 14.60 | **0.93** | 0.02 | 0.80 | 1.03 |
> | | CADS (Ours) | **7.90** | 0.76 | **0.49** | **0.35** | **2.30** |
> | SHHQ | DDIM | 26.27 | **0.70** | 0.15 | 0.57 | 1.27 |
> | | CADS (Ours) | **15.14** | 0.61 | **0.46** | **0.36** | **2.06** |
> | StableDiffusion | DPM++ | 45.70 | **0.70** | 0.29 | 0.19 | 5.30 |
> | | CADS (Ours) | **40.35** | 0.65 | **0.42** | **0.13** | **6.93** |
> | | PNDM | 45.76 | **0.68** | 0.28 | 0.19 | 5.36 |
> | | CADS (Ours) | **41.37** | 0.65 | **0.38** | **0.13** | **6.83** |
>
>
> ### **Confusion about the random seed**
>
> We thank the reviewer for catching this ambiguous terminology. We have changed the term “random seed” to “initial random sample” in our working draft, and we will pay additional attention to this section to ensure clarity when finalizing the manuscript.  The idea we wish to get across is that *low* diversity means that many different initial random samples are converted by the model to a small number of stereotyped outputs, while *high* diversity means that many different initial random samples are converted by the model to *just as many* distinct outputs.  However, when comparing CADS versus non-CADS sampling, we do fix the *local* random seed to guarantee that each method sees the same set of initial random samples, $x_T$.
>
> ### **Higher Inception Score (IS) values for DDPM**
>
> As argued in [3], IS “does not reward covering the whole distribution or capturing diversity within a class.”  Indeed, one can construct a pathological scenario in which a very small set of exemplar images (say, one or a few for each class) can maximize IS while minimizing diversity by design.  Nevertheless, IS is still a popular measure of generative modeling output, which we included for completeness.  As IS is also a reasonable measure of image quality, it is worth noting that it consistently *increases* with $w_{\text{CFG}}$ in Figure 5.  That it is somewhat below DDPM may speak more to the nuances of IS than to the quality of output when using CADS.  Indeed, we report a similar phenomenon in Appendix F, where the IS value computed on *real* data from ImageNet was significantly lower than the values obtained for DDPM samples. As sampling with higher guidance tends to maximize the class probability $p(y|x)$, the IS of DDPM outputs is possibly artificially inflated.
>
> ---
> **References**
>
> [1] Lu, Cheng, et al. "Dpm-solver++: Fast solver for guided sampling of diffusion probabilistic models." *arXiv preprint arXiv:2211.01095* (2022).
>
> [2] Liu, Luping, et al. "Pseudo Numerical Methods for Diffusion Models on Manifolds." *International Conference on Learning Representations*. 2022.
>
> [3] Dhariwal, Prafulla and Nichol, Alex. “Diffusion models beat GANs on image synthesis.” arXiv preprint arXiv:2105.05233v4 (2021).

---

> > ### Comment · Reviewer_iK6t · 2023-11-22
> >
> > Thank you for the clarifications.
> > The authors resolved all of my concerns.
> >
> > I will keep my initial rating.

---

> > > ### Author Response · Authors · 2023-11-22
> > > **Official Response to Reviewer iK6t**
> > >
> > > We wish to extend our thanks to the reviewer once again for the helpful remarks and support of our work.

---

> ### Comment · Area_Chair_94LA · 2023-11-21
>
> Reviewer iK6t,
>
> Please reply to the authors' rebuttal and share your opinion whether it addresses your concerns. Please also post the final decision.
>
> Thanks,
> AC

---

### Official Review · Reviewer_fxon · 2023-11-06

**Soundness:** 3 good
**Presentation:** 4 excellent
**Contribution:** 3 good
**Rating:** 8
**Confidence:** 4

**Summary:**

Although diffusion models are known for good mode coverage, the sample diversity of conditional diffusion models is still challenging when sampling at a high classifier-free guidance (CFG) scale. This work examines the conditional diffusion models and attributes this problem to how the conditional information should be mixed into the reverse diffusion process.

To tackle this problem, it introduces an annealed sampling scheme for conditional generation, Conditional Annealed Diffusion Sampler (CADS). The core principle of CADS is to gradually increase the strength of conditional information in the reverse time diffusion process, letting the unconditional score model explore better data modes in the early stage (noisy region) and guiding the sampling converge to the conditional distribution using the conditional information in the final stage (data region).

This work conducted detailed experiments on class-conditional generation, pose-to-image generation, identity-conditioned face generation, and text-to-image generation. CADS consistently improves the sample diversity of baseline conditional diffusion models without retraining.

**Strengths:**

This is a good paper for conditional diffusion models. Its merits span the following aspects:

1. The proposed method is intuitive and does not require further finetuning on pretrained diffusion models. This allows for improving a wide range of models and samplers. The straightforward approach should be amenable to the broader audience of ICLR.
2. This work also provides a theoretical explanation for CADS. I found this explanation helpful. It should be added to the main paper using the extra granted page after acceptance.
3. Decent experimental results. The proposed sampler offers consistent improvements over the baseline diffusion models.
4. Detailed ablation studies. The ablation studies clearly show the influence of hyperparameters introduced in this sampler.
5. Writing clarity. This paper is well presented. The methods and experiment sections are well organized. Please move Appendix C. to the main article.

**Weaknesses:**

While the paper possesses several strengths, there is room for enhancement in articulating the motivation behind the methodology. The issue of sample diversity has been clearly defined, yet the rationale for the solution could benefit from a stronger motivation. Ideally, the approach should be presented as a natural derivation from the first principle, rather than retroactively justified through theoretical exposition.

This critique should not be seen as a detriment to the overall quality of the work. It is, in essence, a good paper for improving the conditional information in diffusion models.

**Questions:**

1. How does this work determine where to add noises to the conditional information? For example, CADS mostly injects noises into the embeddings. However, for the pose-to-image generation, this work adopts the noise injection to the pose image. Is there a guidance to choose the position for conditional noise injection? What if inject noises into the internal layers of the conditional information extractor? What if we also adopt the embeddings noise injection scheme for the face-to-image generation? In other words, how large can the influence of noise injection position be regarding different choices?
2. Notably, the experiments in this work utilize relatively large sampling steps (>= 100 NFEs). Will the proposed sampler deteriorate at a limited number of function evaluations? How does the conditional generation under CADS change along different sampler steps/NFEs?

---

> ### Author Response · Authors · 2023-11-16
> **Official Response to Reviewer fxon**
>
> ### **Motivating the method**
> We appreciate the reviewer’s suggestion to better motivate the method ahead of its introduction and to move some of the theoretical discussion from the appendix to the main paper.  To the best of our knowledge, accepted papers at ICLR do not get an extra page allowance, so we have to assume for the moment that we are working with limited space.  However, we agree with the reviewer that the placement of the method’s motivation is important, so we have added a paragraph above Section 3.1 in our working draft to provide readers a better intuition behind the rationale of the solution before introducing the solution itself. Please find the corresponding paragraph below:
>
> > We conjecture that the low-diversity issue is due to a highly peaked conditional distribution, particularly at higher guidance scales, which leads the majority of samples toward certain modes during inference. One way to address this issue is by smoothing the conditional distribution with decreasing Gaussian noise, which breaks and then gradually restores the statistical dependency on the conditioning signal during inference. Next, we introduce a novel sampling technique that integrates this intuition into the inference process. The superiority of this method over the above approaches is depicted in Figure 2.c.
>
>
> ### **Where to inject noise**
> Many conditional diffusion models have an embedding layer that provides the actual continuous conditioning signal to the input of the model. This can effectively be a look-up table in the class-conditional case or a continuous functional mapping, such as the CLIP network, for text-to-image generation. This continuous embedding space is ideal for injecting Gaussian noise, and for a more unified approach across various tasks, we injected noise into each condition’s embedding. In some settings, however, such as the pose-to-image task, embedding layers aren’t normally used, as the conditioning information is directly concatenated with the input to the diffusion model [1]. We decided to experiment with the noise injection directly on top of the pose image, since it is this representation that constitutes the conditioning signal in this case. Based on the reviewer’s comment, we also tested a different pose-to-image model that does use an embedding layer and found that injecting noise to the pose embedding also successfully diversifies the model outputs. We saw a similar phenomenon for the face-ID setting, where adding noise either before or after the embedding layer in the diffusion model performed well.  The takeaway seems to be that the technique is highly flexible and robust to this design choice, with the key requirement being initially breaking and then progressively restoring the model’s dependence on the conditioning signal, which can be achieved at a number of places in the model.
>
>
> ### **Behavior of CADS VS NFEs**
> This is an excellent question. We have added a section to the appendix of the paper that explores the effectiveness of CADS with respect to different numbers of sampling steps (NFEs). The behavior of CADS vs NFEs is similar to the base sampler, and a consistent gap exists between sampling with and without condition annealing across different NFEs. Additionally, please note that as DDPM does not perform well when using low NFEs, we switched to DPM++ [2] and DDIM [3] to perform this comparison. Please find the results of this experiment in the tables below.
>
> |  | DPM++ w/ CADS | DPM++ w/ CADS | DPM++ | DPM++ |
> | --- | :---: | :---: | :---: | :---: |
> | **NFEs** | **FID** | **Recall** | **FID** | **Recall** |
> | 25 | 7.73 | 0.67 | 17.86 | 0.37 |
> | 50 | 8.56 | 0.64 | 18.64 | 0.36 |
> | 75 | 8.95 | 0.63 | 18.81 | 0.36 |
> | 100 | 9.63 | 0.61 | 18.90 | 0.36 |
>
> |  | DDIM w/ CADS | DDIM w/ CADS | DDIM | DDIM |
> | --- | :---: | :---: | :---: | :---: |
> | **NFEs** | **FID** | **Recall** | **FID** | **Recall** |
> | 25 | 8.45 | 0.64 | 18.67 | 0.36 |
> | 50 | 9.80 | 0.59 | 18.91 | 0.36 |
> | 75 | 10.08 | 0.57 | 18.92 | 0.35 |
> | 100 | 10.16 | 0.58 | 18.96 | 0.35 |
>
> ---
> **References**
>
> [1] Saharia, Chitwan, et al. "Image super-resolution via iterative refinement (2021)." *arXiv preprint arXiv:2104.07636* (2021).
>
> [2] Lu, Cheng, et al. "Dpm-solver++: Fast solver for guided sampling of diffusion probabilistic models." *arXiv preprint arXiv:2211.01095* (2022).
>
> [3] Song, Jiaming, Chenlin Meng, and Stefano Ermon. "Denoising Diffusion Implicit Models." *International Conference on Learning Representations*. 2021.

---

> > ### Comment · Reviewer_fxon · 2023-11-20
> >
> > I really appreciate the author's timely response. This perfectly addressed my questions. Very interestingly, the proposed CADS works even better at small NFEs.
> >
> > - Could you please further reduce the NFEs? Given the current observations, I assume there will be a U curve regarding FIDs.
> >
> > - In addition, which dataset/setup was used for this table? (It would be better if this could be further verified using additional setups/models once authors have more time after acceptance. There's no rush to complete it now. Please take your time with this, considering the limited discussion time remaining.)
> >
> > It is a great pleasure to review this paper. I have no concerns about this work and will be very happy to see the acceptance.
> >
> > Thanks,\
> > Reviewer fxon

---

> > > ### Author Response · Authors · 2023-11-22
> > > **Official Response to Reviewer fxon**
> > >
> > > We would like to thank the reviewer once again for the helpful and supportive comments.
> > >
> > > The NFE experiment we reported is based on class-conditional ImageNet generation using the DiT-XL/2 model. The reviewer’s intuition is indeed correct that reducing NFEs further would result in a U-shaped curve in the FID. We have performed follow-up tests at lower NFEs and have observed the U-shaped behavior that the reviewer suggested. We will report these results in the final version of the paper and would also be happy to extend this experiment beyond the ImageNet generation task.

---

### Author Response · Authors · 2023-11-16
**Thanks to all reviewers**

We thank all reviewers for their time, helpful comments, and positive reaction to our work. We have prepared responses to individual reviewers’ comments and questions, and we welcome any follow-up discussion.

---

### Meta-Review · Area_Chair_94LA · 2023-12-09

**Metareview:**

The paper introduces the Condition-Annealed Diffusion Sampler (CADS), a novel strategy to enhance the diversity of outputs in conditional diffusion models, especially at high guidance scales, while maintaining sample quality, by modulating the conditioning signal during inference. All the reviewers recommend acceptance. They all agree that the paper proposes a solid approach the experiments are strong. After carefully reading the rebuttal and paper, the AC agrees with the reviewers on accepting the paper.

**Justification For Why Not Higher Score:**

The image quality looks reasonable but not as good as shown in state-of-the-art public diffusion models.

**Justification For Why Not Lower Score:**

The paper provides a solid approach and experiments.

---

### Decision · Program_Chairs · 2024-01-16

Accept (spotlight)